# Rethinking Instruction Drift as a Sampling Error: SNR-Aware Power Distributions for Long-Horizon Robotic Planning

**Kewei Chen** [1 2]   **Yayu Long** [1 2]   **Mingsheng Shang** [1 2]

## Abstract

Despite rapid progress in Vision-Language-Action (VLA) models for robotic control, instruction drift remains a persistent failure mode in long-horizon tasks. This paper reconceptualizes this phenomenon, positing that instruction drift is fundamentally a systematic sampling error: local greedy sampling is prone to collapsing into "Negative Pivotal Windows"—irreversible local optima with high local probability that sever global success pathways. To address this, we propose **Context-Aware Power Sampling (CAPS)**, a training-free inference-time computation framework. CAPS leverages power distributions to sharpen global trajectory probabilities, enabling lookahead search over the model's conditional generative trajectory distribution. Furthermore, we introduce a metacognitive control mechanism based on Signal-to-Noise Ratio (SNR). This mechanism triggers adaptive MCMC search solely when drift risk is detected, enabling a dynamic transition from "intuitive fast thinking" to "rational slow search." Experiments on RoboTwin, Simpler-WindowX, and Libero-long benchmarks show that CAPS achieves substantial improvements over strong baselines, including OpenVLA and TACO, without parameter updates. These results support the effectiveness of adaptive inference-time computation for improving long-horizon robustness in embodied control.

## 1. Introduction

In recent years, Vision-Language-Action (VLA) models have emerged as **core enablers** for general robotic control (Zitkovich et al., 2023; Kim et al., 2025). While these models excel in short-horizon tasks, they frequently encounter critical failure modes in complex, long-horizon scenarios, specifically **Instruction Drift** (Zhang et al., 2025b; Ouyang et al., 2025). As tasks progress, irrelevant observations—such as background noise or non-target object movements—gradually dilute the attention weights assigned to the initial instruction. Consequently, the robot loses contextual grounding with the ultimate goal, executing actions that are locally reasonable but globally erroneous (Dasari et al., 2025; Huang et al., 2025; Zhang et al., 2025a).

Existing solutions primarily follow two paths: prompt engineering and inference-time scaling. Prompt engineering, such as Chain-of-Thought (CoT), attempts to enhance logical coherence but remains constrained by the unidirectional nature of open-loop generation (Shen et al., 2025; Zhao et al., 2025). Conversely, methods like TACO (Yang et al., 2025) adopt a "generate-verify" paradigm, modeling inference as a contextual bandit problem. However, these approaches typically rely on parallel independent sampling and reranking. They lack the capacity for *Iterative Refinement* of a single trajectory (Yan et al., 2025). Once the candidate set lacks high-quality solutions, simple reranking becomes ineffective.

To address these challenges, we propose a novel theoretical perspective: the root cause of instruction drift lies in systematic biases within sampling strategies. Specifically, existing local greedy strategies suffer from the myopia of single-step prediction (Zhang et al., 2026; Simchowitz et al., 2025). The resulting trajectories, while locally smooth in the time domain, often fracture in global physical consistency. Based on this perspective, **Inference-time Scaling** (Wu et al., 2025; Pan et al., 2026) provides a promising framework for mitigating such systematic errors at the distribution level. While this paradigm has achieved success in discrete symbolic reasoning tasks like mathematics (Karan & Du, 2026), its application in continuous visuo-motor control remains underexplored. This lag stems from a fundamental *Topological Mismatch*: unlike discrete token generation, robotic policies must operate within high-dimensional continuous action manifolds governed by rigorous, irreversible physical consequences. This makes unconstrained exhaus-

[1]Chongqing Institute of Green and Intelligent Technology, Chinese Academy of Sciences [2]Chongqing School, University of Chinese Academy of Sciences. Correspondence to: Mingsheng Shang <msshang@cigit.ac.cn>.

*Proceedings of the 43rd International Conference on Machine Learning*, Seoul, South Korea. PMLR 306, 2026. Copyright 2026 by the author(s).

tive sampling—often feasible in discrete domains—difficult to apply under the real-time constraints of closed-loop control. This topological mismatch makes greedy-policy-based robots highly prone to collapsing into "Negative Pivotal Windows"—irreversible local optima that possess high local probability but physically sever all paths to global success.

To bridge this gap, we propose **Context-Aware Power Sampling (CAPS)** (see Figure 1). As a dual-process architecture designed for continuous control, CAPS reformulates inference as constrained search on the action manifold under the model's conditional generative trajectory distribution. The framework utilizes *Information-Theoretic Criticality* to dynamically identify *Topological Bifurcation Points* in the action space (Liu et al., 2025b). The system seamlessly transitions from greedy execution (System 1) to global optimization (System 2) only when decision uncertainty spikes. Within System 2, we introduce a *Randomized Resampling Proposal* to escape local optima and enforce long-horizon physical consistency via a Power Distribution-based Metropolis-Hastings *Acceptance Check*. This mechanism performs *Iterative Rectification* on trajectories, improving robustness while preserving practical inference efficiency. Experimental results on RoboTwin, Simpler-WindowX, and Libero-long benchmarks show that CAPS demonstrates clear advantages over OpenVLA (Kim et al., 2025) and TACO (Yang et al., 2025) in a training-free setting, supporting the effectiveness of shifting computational resources from training to inference for long-horizon robustness.

The main contributions of this paper are:

(1)**Theoretical Reframing:** We reinterpret long-horizon robotic instruction drift through the lens of sampling error. We introduce global power distributions as a mathematical framework for analyzing this phenomenon and contrast them with traditional local temperature sampling.

(2)**CAPS Framework:** We propose CAPS, which uses SNR-modulated adaptive MCMC to allocate inference-time compute dynamically. This instantiates a "search-when-necessary" strategy for robotic control.

(3)**Strong Empirical Performance:** On long-horizon benchmarks including RoboTwin, Simpler-WindowX, and Libero-long, CAPS achieves strong training-free performance and demonstrates clear advantages over OpenVLA and TACO. These results support the effectiveness of inference-time compute for improving long-horizon consistency.

**Conflict of Interest Disclosure.** The authors declare no financial conflicts of interest related to this work.

## 2. Related Work

**Robotic Instruction Following and Inference-Time Scaling.** Large Language Models (LLMs) and Vision-Language-Action (VLA) models have become central drivers of robotic planning (Zitkovich et al., 2023; Kim et al., 2025; Ichter et al., 2023). To address uncertainty in long-horizon tasks, TACO (Yang et al., 2025) models inference as a contextual bandit problem, utilizing a verifier to re-rank parallel candidate actions; Chain-of-Thought (CoT) (Wei et al., 2022) and ReAct (Yao et al., 2023b) focus on inducing intermediate reasoning steps. However, the former is limited by the quality of the candidate set, while the latter remains essentially open-loop generation. In contrast, CAPS discards external verifiers and parallel sampling, instead employing an intra-inference closed-loop mechanism to directly optimize single trajectory generation quality, thereby more efficiently correcting long-horizon drift.

**Inference-Time Computation and Sampling.** Inference-time compute has emerged as an important paradigm for enhancing complex reasoning capabilities (Yao et al., 2023a; Lightman et al., 2024). However, existing discrete sampling strategies face a dilemma in long-horizon embodied control. Traditional low-temperature sampling is inherently *myopic* (Fu et al., 2024; Karan & Du, 2026), causing policies to fall into local optima. While Tree of Thoughts (Yao et al., 2023a) and RoboMonkey (Kwok et al., 2025) introduce global search, they often rely on fixed, high computational budgets, making them ill-suited for the real-time constraints of continuous action manifolds. CAPS adopts a metacognitive "Search-when-necessary" strategy that aims to mitigate this topological mismatch through SNR-based dynamic compute allocation.

**Power Distributions and Energy Models.** From a statistical physics perspective, (Faria et al., 2024) utilized Metropolis-Hastings sampling in machine translation to approximate quality-weighted distributions. (Karan & Du, 2026) further proved that sampling from a power distribution $\pi(\tau) \propto p(\tau)^{\alpha}$ is equivalent to implicit planning over future trajectories. This paper extends this line of theory to the domain of embodied intelligence. We propose using global power distributions to sharpen the probability landscape, offering a theoretical perspective for understanding and mitigating instruction drift in continuous action spaces.

## 3. Methodology

This section details **Context-Aware Power Sampling (CAPS)**, an inference-time adaptive computation framework designed for continuous control. CAPS aims to address the Manifold Alignment problem in high-dimensional action spaces by constructing a dual-process control architecture that establishes a dynamic balance between fast

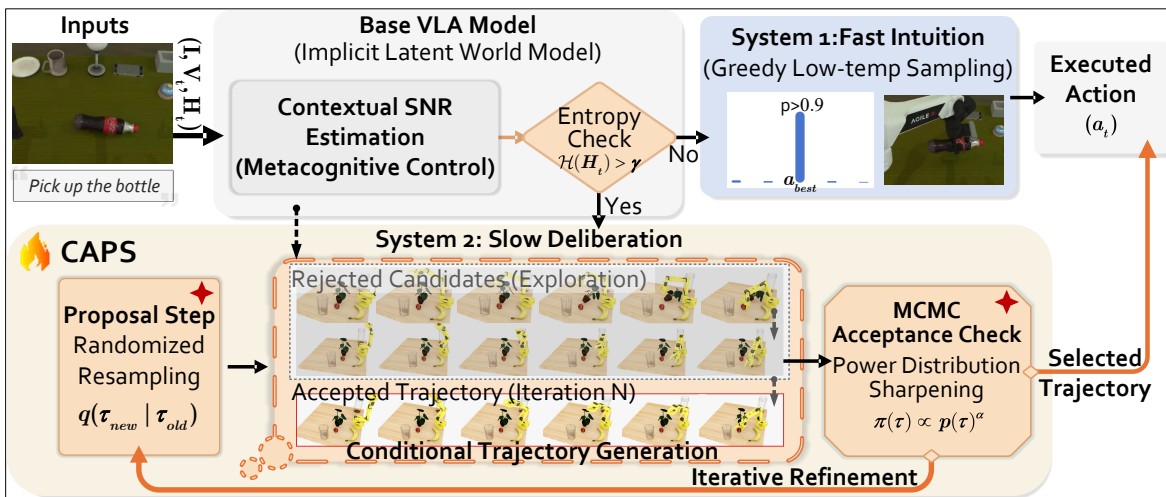

*Figure 1.* **Overview of the Context-Aware Power Sampling (CAPS) framework.** The system processes inputs $(I, V_t, H_t)$ and employs Metacognitive Control to dynamically gate computation based on contextual SNR. In high-certainty scenarios ($\mathcal{H} \leq \gamma$), it executes fast greedy sampling (**System 1: Fast Intuition**). Conversely, when an entropy spike is detected ($\mathcal{H} > \gamma$), it activates **System 2: Slow Deliberation**. As illustrated in the conditional trajectory generation and verification loop, the planner performs a lookahead search: it iteratively generates proposals via randomized resampling ($q(\tau_{new}|\tau_{old})$), filtering out rejected candidates while retaining the accepted trajectory that satisfies the sharpened power distribution ($\pi \propto p^\alpha$). This Iterative Refinement mechanism effectively converts inference-time compute into long-horizon robustness.

heuristic execution and slow global optimization. The framework samples and refines candidate future trajectories under the model's conditional generative trajectory distribution, thereby correcting instruction drift caused by local horizons in long-horizon planning tasks.

### 3.1. Problem Definition: From Text Generation to Trajectory-Level Search

In long-horizon robotic control tasks, we formalize the problem as generating an action sequence $\tau$ given natural language instructions $I$ and the current observation history $H_t$. Traditional autoregressive generation $p_\theta(\tau|I, H_t)$ typically employs greedy sampling, which often falls into local optima when handling complex long-horizon dependencies due to ignoring the *Global Topology of Action Manifold*. Therefore, we reframe the inference objective as sampling from a global **Power Distribution**:

$$\pi(\tau) \propto p_\theta(\tau|I, H_t)^\alpha, \quad \alpha \geq 1 \quad (1)$$

where $p_\theta$ represents the base model's original probability distribution, and $\alpha$ is the sharpening coefficient. This distribution, via mathematical sharpening, reweights the model's conditional generative trajectory distribution toward trajectories with higher long-horizon consistency, thereby avoiding negative pivotal windows where local probability is high but long-term feasibility is absent.

### 3.2. Adaptive Computation as Optimal Control

To balance inference precision and computational budget, we model the selection of the sampling strategy as a resource-constrained sequence decision problem. The switching mechanism of CAPS is based on the optimization of a computational utility function, dynamically allocating computing resources according to the current contextual uncertainty.

#### 3.2.1. METACOGNITIVE SIGNAL: KL DIVERGENCE-BASED SNR

To quantify signal decay during long-horizon planning, we view instruction drift as a process where the policy distribution gradually degrades into random noise. Therefore, we define **Contextual SNR** as the information gain of the current policy $\pi_\theta$ relative to the **Maximum Entropy Noise Floor**.

Formally, we use Kullback-Leibler (KL) divergence to measure this "distance between dominant signal and background noise":

$$\text{SNR}_t \triangleq D_{\text{KL}}(\pi_\theta(\cdot|H_t) \,\|\, \mathcal{U}_{\text{unif}}) = \sum_{a \in \mathcal{A}} \pi_\theta(a|H_t) \log \frac{\pi_\theta(a|H_t)}{1/|\mathcal{A}|} \quad (2)$$

where $\mathcal{U}_{\text{unif}}$ is the uniform distribution over the action space $\mathcal{A}$ (representing pure noise). Expanding the KL divergence term yields:

$$\text{SNR}_t = \log |\mathcal{A}| - \mathcal{H}(\pi_\theta(\cdot|H_t)) \quad (3)$$

Since the action space size $|\mathcal{A}|$ is constant during infer-

ence, Equation (3) indicates that $\text{SNR}_t$ exhibits a strict linear negative correlation with the policy's Shannon entropy $\mathcal{H}(\pi_\theta)$. From a topological geometry perspective, a sharp increase in entropy often corresponds precisely to "Bifurcation Points" on the probability manifold—critical moments where the model faces choices among multiple potential paths. This perspective provides a solid theoretical basis for using entropy as an efficient proxy metric for drift detection: when $\mathcal{H}(\pi_\theta)$ rises, it directly implies that the effective signal $\text{SNR}_t$ is being drowned out by background noise, necessitating the activation of System 2 for correction.

**Optimal Switching Policy.** Our goal is to minimize the total expected loss $\mathcal{L}(\pi) = \mathbb{E}[\text{Error}|\pi] + \lambda \cdot \mathcal{C}(\pi)$, where $\mathcal{C}(\pi)$ represents computational cost, and $\lambda$ is the Lagrange multiplier regulating the trade-off between precision and budget. Under this formulation, the resulting policy takes the form of a hard-threshold switching rule (derivation in Appendix F):

$$
\pi^*(H_t) = \begin{cases} \pi_{\text{slow}}(\text{CAPS}) & \text{if } \mathcal{H}(\pi_\theta(\cdot|H_t)) > \gamma(\lambda) \\ \pi_{\text{fast}}(\text{Greedy}) & \text{otherwise} \end{cases} \quad (4)
$$

where $\gamma(\lambda)$ is the entropy threshold implicitly determined by $\lambda$. This suggests that the system should allocate additional computation only when uncertainty exceeds the tolerance limit determined by the budget (i.e., high-entropy pivotal windows), yielding an efficient trade-off between robustness and computation.

### 3.2.2. BLOCK-BASED AUTOREGRESSIVE MCMC (INTRA-INFERENCE LOOP)

When the system switches to $\pi_{\text{slow}}$ mode, we utilize the Metropolis-Hastings algorithm to construct an intra-inference closed-loop correction process. Figure 2 visually illustrates this iterative refinement mechanism composed of Randomized Resampling (Proposal) and Global Sharpening Verification (Acceptance). The process consists of three core stages:

**Proposal.** We adopt a randomized resampling strategy. For mathematical rigor, we explicitly define the proposal distribution $q(\tau_{new}|\tau_{old})$ as the conditional generation distribution of the base model $p_\theta$ under the current context. Specifically, we keep the prefix of the current action block unchanged and resample the suffix using $p_\theta$ with a temperature of $1/\alpha$ to generate a candidate trajectory $\tau_{new}$. This step explores potential alternative paths by resampling from the model's conditional generative trajectory distribution within the local neighborhood of the Action Manifold.

**Acceptance.** The system decides whether to accept the candidate correction based on the likelihood ratio under the power distribution. The acceptance rate $A(\tau_{old}, \tau_{new})$ is

defined as:

$$
A(\tau_{old}, \tau_{new}) = \min\left(1, \frac{p_\theta(\tau_{new}|I, H_t)^\alpha}{p_\theta(\tau_{old}|I, H_t)^\alpha} \cdot \frac{q(\tau_{old}|\tau_{new})}{q(\tau_{new}|\tau_{old})}\right) \quad (5)
$$

The first term represents the likelihood contrast of the two trajectories under the global power distribution, while the second term is the correction factor for the proposal distribution (Hastings Correction). A high $\alpha$ value acts as a strict verifier, ensuring that only trajectories highly consistent with instruction $I$ and history $H_t$ are retained.

**Iterative Refinement.** The proposal and acceptance process described above is repeated $N_{\text{MCMC}}$ times. This iteration mechanism ensures that the system can fully explore and verify candidate trajectories within the internal probability space before outputting actions. This shifts the inference process from flat generation toward structured cognitive control.

### 3.2.3. CHUNK-LEVEL APPROXIMATION AND LONG-HORIZON INTERPRETATION

Eq. (1) defines the target distribution at the trajectory level, whereas the practical implementation of CAPS operates on finite action blocks. Directly optimizing over the entire future trajectory is computationally prohibitive at inference time. Thus, CAPS decomposes the global search problem into an autoregressive sequence of local block refinements. Analogous to receding-horizon control, this block-based MCMC serves as a computationally tractable approximation, providing a practical mechanism to incrementally maximize the joint trajectory probability under real-time constraints.

The SNR gating mechanism links these local interventions to the horizon-level theory. By triggering MCMC refinement only at low-SNR decision points (high-uncertainty pivotal windows), CAPS selectively targets the localized failures that most strongly dictate long-range error accumulation. In this context, Theorem 3.1 provides the idealized trajectory-level perspective, while Theorem 3.2 characterizes how finite MCMC steps on local chunks influence the practical horizon extension.

Finally, despite operating on local blocks, the refinement process retains a global lookahead property. The acceptance step evaluates candidate chunks under the sharpened conditional generation distribution induced by the base VLA model, which implicitly encodes long-horizon priors learned during pre-training. Consequently, while CAPS avoids explicit full-sequence unrolling, it still scores and selects local actions according to their compatibility with broader long-term trajectory consistency.

**Theoretical Analysis and Physical Interpretation.** To explain the efficacy of CAPS mechanistically, we explore in **Appendix G** how CAPS reshapes the energy landscape to

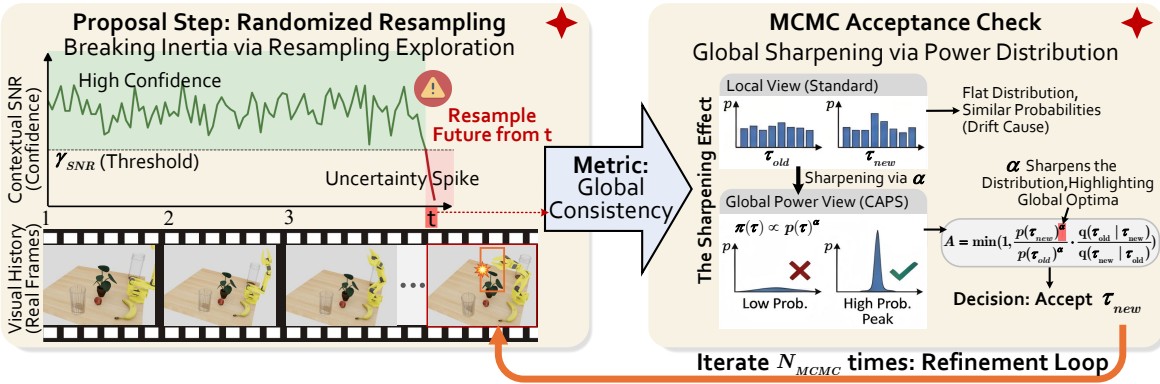

*Figure 2.* **Mechanism of the System 2 Inference Loop. Left (Proposal):** Triggered by an SNR drop below the critical threshold $\gamma_{SNR}$ at time $t$, the system breaks inertia by initiating stochastic resampling of future trajectories. **Right (Acceptance):** The proposed candidate ($\tau_{new}$) is compared against the incumbent ($\tau_{old}$) via a global consistency metric. By applying power distribution sharpening ($\alpha$), CAPS amplifies probability gaps to robustly identify global optima. This refine-and-check loop iterates $N_{MCMC}$ times, effectively converting inference-time compute into trajectory robustness.

avoid local minima from the perspective of Energy-Based Models (EBM). We also discuss the mechanism's dependence on model calibration. Furthermore, in **Appendix H**, we provide an analysis of how global sharpening may extend the robot's *Effective Horizon* from linear to polynomial order under the assumptions of our model, offering theoretical support for the performance trends observed in long-horizon tasks.

**Algorithm Implementation.** To clearly demonstrate how CAPS dynamically coordinates "Fast Intuition" and "Slow Search" at inference time, we provide the complete pseudocode in **Appendix O**.

### 3.3. Theoretical Insight: Effective Horizon Extension

To quantify the benefits of CAPS in long-horizon tasks, we introduce the concept of **Effective Horizon** ($T_{eff}$), defined as the maximum number of steps a policy can sustain while maintaining a success rate confidence $\eta$.

**Theorem 3.1** (Horizon Extension Theorem - Ideal Bound). *Assume the base policy has a single-step drift probability $\epsilon$ at pivotal windows. Under the ideal sampling assumption, CAPS global sharpening ($\alpha > 1$) extends the effective horizon from linear to polynomial order:*

$$\frac{T_{eff}(CAPS)}{T_{eff}(Base)} \approx \epsilon^{1-\alpha} \qquad (6)$$

However, actual inference is constrained by the computational budget. Considering the convergence process of MCMC, we derive the asymptotic lower bound under finite steps $N$.

**Theorem 3.2** (Asymptotic Horizon Extension - Performance Lower Bound). *Assume the base policy has a single-step drift rate $\epsilon$. Under MCMC sampling with finite steps $N$,*

*the effective horizon extension multiple of CAPS satisfies:*

$$\frac{T_{eff}(CAPS)}{T_{eff}(Base)} \approx \frac{\epsilon}{\epsilon^{\alpha} + \underbrace{\mathcal{O}(\rho^N)}_{\text{Sampling Bias}}} \xrightarrow{N \to \infty} \epsilon^{1-\alpha} \qquad (7)$$

*where $\rho \in (0,1)$ is the mixing rate of the Markov chain, and $\mathcal{O}(\rho^N)$ represents the residual sampling error due to incomplete MCMC convergence.*

**Physical Significance:** Under the assumptions of our analysis, Theorem 3.1 suggests that global sharpening can substantially suppress the effective drift rate. For example, setting $\alpha = 2$ and $\epsilon = 0.1$ yields a $10\times$ horizon extension in the idealized setting. Theorem 3.2 further characterizes how this effect depends on the available computational budget and the residual sampling error $\mathcal{O}(\rho^N)$. Together, these results offer a plausible explanation for the long-horizon robustness trends observed in our experiments.

## 4. Experiments

We evaluate **CAPS** on three benchmarks: RoboTwin (Mu et al., 2025), Simpler-WindowX (Li et al., 2025), and Libero-long (Liu et al., 2023), using 4 NVIDIA A100 GPUs (config in Appendix M). We address: **1. Robustness** in complex manipulation; **2. OOD Generalization**; and **3. Long-Horizon Consistency** against reranking baselines.

### 4.1. RoboTwin Benchmark: Bimanual Collaboration and Complex Manipulation

RoboTwin tests precise bimanual coordination where minor errors cause total failure. We evaluated CAPS on RoboTwin 1.0 with $\pi_0$. Table 1 shows CAPS improves $\pi_0$ to 47.4% (+15.2%), surpassing TACO (+6.1%). This suggests an advantage of active iterative refinement (MCMC) over pas-

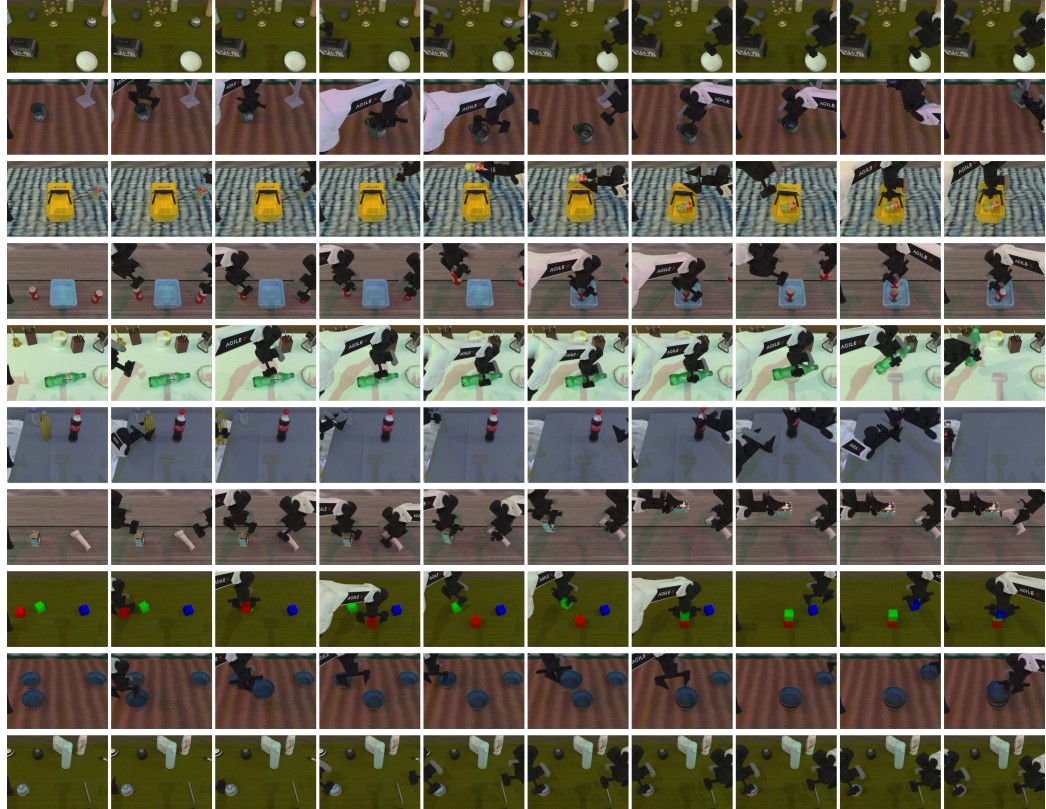

*Figure 3.* **Visualizing Bimanual Complexity on RoboTwin 2.0.** The figure displays execution snapshots of CAPS in various high-difficulty bimanual tasks. These tasks require extreme spatiotemporal coordination. CAPS effectively suppresses local drift of a single arm through global sharpening, significantly improving the success rate of bimanual collaboration.

*Table 1.* Evaluation of success rate(%) on the RoboTwin 1.0.

| | Block Handover | Bottles Adjust | Container Place | Diverse Bottles Pick | Dual Bottles Pick Easy |
|---|---|---|---|---|---|
| $\pi_0$ (Black et al., 2024) | 41.0 | 31.0 | 25.0 | 21.0 | 60.0 |
| $\pi_0$ + TACO (Yang et al., 2025) | 62.0 | 40.0 | 40.0 | 27.0 | 70.0 |
| **CAPS(Ours)** | **67.0** | **51.0** | **49.0** | **31.0** | **72.0** |
| | Dual Bottles Pick Hard | Pick Apple Messy | Shoe Place | Mug Hanging Easy | Average |
| $\pi_0$ (Black et al., 2024) | 48.0 | 15.0 | 42.0 | 7.0 | 32.2 |
| $\pi_0$ + TACO (Yang et al., 2025) | 52.0 | 19.0 | **50.0** | 12.0 | 41.3 |
| **CAPS(Ours)** | **61.0** | **26.0** | 49.0 | **21.0** | **47.4** |

sive screening. In the drift-prone "Dual Bottles Pick Hard," CAPS raised success from 48.0% to 61.0% via iterative search.

### 4.2. Simpler-WindowX: OOD Generalization

Simpler-WindowX evaluates robustness against visual changes (low SNR scenarios). As shown in Table 2, CAPS achieved 60.5% with $\pi_0$, significantly outperforming specialized models like SpatialVLA (42.7%). In high-interference tasks like "Carrot on Plate" (+19.0%), the SNR gating mechanism effectively triggers resampling when noise lowers confidence, planning robust paths in low-SNR regions.

### 4.3. RoboTwin 2.0: Large-Scale Test-Time Scaling

Results on RoboTwin 2.0 (Table 3) confirm scalability. CAPS achieved optimal performance (66.2%), outperforming RDT (34.6%) and TACO (64.0%). Notably, in long-horizon tasks like "Lift Pot," CAPS improved by 14.0% over $\pi_{0.5}$, showing that global power distribution sampling effectively suppresses error accumulation. In simple tasks, CAPS maintains parity with baselines, validating the efficiency of SNR gating in avoiding overhead. Execution snapshots are in Figure 3.

### 4.4. Libero-long: Long-Horizon Tasks

Libero-long tests instruction consistency. Table 4 shows CAPS achieved 97.6% success, outperforming OpenVLA

*Table 2.* Success rate (%) evaluated on Simpler-WindowX. Baseline results are taken from *Simpler*, *RoboVLM*, and *SpatialVLA*. Our method achieves an average improvement of 12.5% over $\pi_0$.

| | RT-1-X (O'Neill et al., 2024) | Octo (Mees et al., 2024) | RoboVLM (Li et al., 2026) | SpatialVLA (Qu et al., 2025) | $\pi_0$ (Black et al., 2024) | $\pi_0$ + TACO (Yang et al., 2025) | **CAPS** **(Ours)** |
|---|---|---|---|---|---|---|---|
| Spoon on Towel | 0.0 | 12.5 | 29.2 | 16.7 | 36.0 | 52.0 | **57.0** |
| Carrot on Plate | 4.2 | 8.3 | 25.0 | 25.0 | 42.0 | 52.0 | **61.0** |
| Stack Cubes | 0.0 | 0.0 | 12.5 | 29.2 | 34.0 | 30.0 | **37.0** |
| Eggplant in Basket | 0.0 | 43.1 | 58.3 | **100.0** | 80.0 | 88.0 | 87.0 |
| Average | 1.1 | 16.0 | 31.3 | 42.7 | 48.0 | 55.5 | **60.5** |

*Table 3.* RoboTwin 2.0 Benchmark Evaluation. Each task is tested using 100 different seeds across 100 randomly generated scenes. For test-time scaling, the number of candidate actions is set to 50.

| | Adjust Bottle | Beat Block Hammer | Blocks Ranking Size | Dump Bin Bigbin | Grab Roller | Handover Block | Handover Mic |
|---|---|---|---|---|---|---|---|
| RDT (Liu et al., 2025a) | 81.0 | 77.0 | 0.0 | 64.0 | 74.0 | **45.0** | **90.0** |
| $\pi_{0.5}$ (Black et al., 2025) | 89.0 | 69.0 | 36.0 | 86.0 | **100.0** | 24.0 | 58.0 |
| $\pi_{0.5}$ + TACO | 93.0 | 79.0 | 40.0 | 87.0 | 98.0 | 36.0 | 63.0 |
| **CAPS (Ours)** | **95.0** | **81.0** | **46.0** | **89.0** | 94.0 | 38.0 | 72.0 |

| | Lift Pot | Move Can Pot | Move Pillbottle Pad | Move Playingcard Away | Move Stapler Pad | Open Laptop | Open Microwave |
|---|---|---|---|---|---|---|---|
| RDT (Liu et al., 2025a) | 72.0 | 25.0 | 8.0 | 43.0 | 2.0 | 59.0 | **37.0** |
| $\pi_{0.5}$ (Black et al., 2025) | 62.0 | 42.0 | **55.0** | 85.0 | 13.0 | 67.0 | 20.0 |
| $\pi_{0.5}$ + TACO | 66.0 | 57.0 | 54.0 | **88.0** | 18.0 | 67.0 | 21.0 |
| **CAPS (Ours)** | **76.0** | **61.0** | 53.0 | 72.0 | **21.0** | **74.0** | 27.0 |

| | Pick Diverse Bottles | Pick Dual Bottles | Place A2B Left | Place A2B Right | Place Bread Basket | Place Bread Skillet | Place Burger Fries |
|---|---|---|---|---|---|---|---|
| RDT (Liu et al., 2025a) | 2.0 | 42.0 | 3.0 | 1.0 | 10.0 | 5.0 | 50.0 |
| $\pi_{0.5}$ (Black et al., 2025) | 52.0 | 72.0 | 51.0 | 39.0 | 62.0 | 62.0 | 89.0 |
| $\pi_{0.5}$ + TACO | 59.0 | **73.0** | **56.0** | 42.0 | **65.0** | 61.0 | 92.0 |
| **CAPS (Ours)** | **62.0** | 71.0 | 54.0 | **47.0** | 63.0 | **69.0** | **93.0** |

| | Place Cans Plasticbox | Place Container Plate | Place Dual Shoes | Place Fan | Place Mouse Pad | Place Object Basket | Place Object Scale |
|---|---|---|---|---|---|---|---|
| RDT (Liu et al., 2025a) | 6.0 | 78.0 | 4.0 | 12.0 | 1.0 | 33.0 | 1.0 |
| $\pi_{0.5}$ (Black et al., 2025) | 68.0 | 85.0 | 23.0 | 34.0 | 16.0 | 69.0 | 45.0 |
| $\pi_{0.5}$ + TACO | 72.0 | **87.0** | 28.0 | 32.0 | 24.0 | **78.0** | 52.0 |
| **CAPS (Ours)** | **79.0** | **87.0** | **33.0** | **36.0** | **25.0** | 76.0 | **57.0** |

| | Place Object Stand | Place Phone Stand | Place Shoe | Put Object Cabinet | Rotate QRcode | Shake Bottle Horizontally | Shake Bottle |
|---|---|---|---|---|---|---|---|
| RDT (Liu et al., 2025a) | 15.0 | 15.0 | 35.0 | 33.0 | 50.0 | 84.0 | 74.0 |
| $\pi_{0.5}$ (Black et al., 2025) | 68.0 | 76.0 | 53.0 | 54.0 | 62.0 | **100.0** | 98.0 |
| $\pi_{0.5}$ + TACO | 78.0 | 86.0 | 65.0 | 56.0 | 67.0 | **100.0** | **99.0** |
| **CAPS (Ours)** | **81.0** | **89.0** | **67.0** | **58.0** | **69.0** | **100.0** | 98.0 |

| | Stack Blocks Three | Stack Blocks Two | Stack Bowls Three | Stack Bowls Two | Stamp Seal | Turn Switch | Average |
|---|---|---|---|---|---|---|---|
| RDT (Liu et al., 2025a) | 2.0 | 21.0 | 51.0 | 76.0 | 1.0 | 35.0 | 34.6 |
| $\pi_{0.5}$ (Black et al., 2025) | 43.0 | 81.0 | 64.0 | 93.0 | 26.0 | 42.0 | 59.3 |
| $\pi_{0.5}$ + TACO | 45.0 | **91.0** | 68.0 | 93.0 | 38.0 | **49.0** | 64.0 |
| **CAPS (Ours)** | **52.0** | 89.0 | **74.0** | **97.0** | **42.0** | 47.0 | **66.2** |

(49.8%) and RoboMonkey (56.5%). In "Moka Pots on Stove," CAPS improved success from 68.0% to 91.0%, indicating MCMC search effectively avoids local optima at Negative Pivotal Windows. Detailed execution sequences and qualitative visualizations of CAPS on Libero-long are provided in Appendix N.

### 4.5. Real-World Evaluation

To rigorously verify the Sim-to-Real Transferability of CAPS, we established a comprehensive benchmark suite containing **10 long-horizon tasks** (see Appendix L for the full list and detailed protocol). Figure 5 (in Appendix) displays the qualitative execution process of representative bimanual collaboration and long-sequence tasks. These tasks cover bimanual coordination, sequence planning, and fine manipulation under unstructured visual noise.

**Setup and Protocol.** Experiments were conducted on the XLeRobot platform using $\pi_0$ as the base model. To ensure statistical significance, we performed a total of 200 trials (10 tasks × 20 trials). In this section, we focus on two representative case studies. Quantitative results for all 10 tasks are summarized in Table 6, with detailed data in the Appendix.

**Case Study 1: Collaborative Storage (Context Retention).** The robot must lift a lid with its left hand and hold it while placing fruit inside with its right hand. The base policy often releases the left hand prematurely (Drift) due to an inability to handle parallel subgoals. CAPS triggers MCMC search upon detecting the high-entropy state and successfully maintains the "holding" state.

**Case Study 2: Multi-stage Breakfast Preparation (Long-Horizon Memory).** The task requires placing bread followed by a cup. The base model often stalls after step one. CAPS successfully identifies the low SNR signal of the uncompleted instruction and plans the subsequent actions.

**Overall Performance.** As shown in Table 6, CAPS

*Table 4.* Evaluation results on the Libero-long benchmark. We apply TACO and CAPS to $\pi_{0.5}$. For the autoregressive VLA architecture, the action sampling temperature is set to 1. Results marked with * are directly cited from Robomonkey (Kwok et al., 2025).

| | $\pi_{0.5}$ (Black et al., 2025) | + TACO (Yang et al., 2025) | **CAPS** **(Ours)** | OpenVLA* (Kim et al., 2025) | Robomonkey* (Kwok et al., 2025) |
|---|---|---|---|---|---|
| Soup and Sauce in Basket | 98.0 | **100.0** | **100.0** | 36.0 | 59.0 |
| Cheese and Butter in Basket | **100.0** | 96.0 | 98.0 | 70.0 | 79.0 |
| Turn on Stove and Place Moka | 98.0 | 98.0 | **99.0** | 58.0 | 58.0 |
| Black Bowl in Drawer | 98.0 | **100.0** | 99.0 | 36.0 | 37.0 |
| Mugs on Plates | **98.0** | 98.0 | 98.0 | 42.0 | 55.0 |
| Book in Caddy | **100.0** | **100.0** | **100.0** | 84.0 | 86.0 |
| Mug and Pudding on Plate | 96.0 | 92.0 | **97.0** | 48.0 | 59.0 |
| Soup and Cheese in Basket | 94.0 | **100.0** | 99.0 | 56.0 | 62.0 |
| Moka Pots on Stove | 68.0 | 86.0 | **91.0** | 26.0 | 26.0 |
| Mug in Microwave | **98.0** | 96.0 | 95.0 | 42.0 | 44.0 |
| Average | 94.8 | 96.6 | **97.6** | 49.8 | 56.5 |

*Table 5.* **Ablation results on RoboTwin 1.0.** We compare CAPS against three variants to isolate the impact of sharpening, adaptive gating, and the search algorithm. Latency is relative to the base policy $\pi_0$.

| Method / Variant | Mechanism Alteration | Success (%) | Latency ($\times$) |
|---|---|---|---|
| **Base Policy ($\pi_0$)** | Greedy / Low-temp Sampling | 32.2 | 1.00 |
| (a) w/o Sharpening | $\alpha = 1$ (Flat Distribution Search) | 38.5 | 8.50 |
| (b) Rejection Sampling | MCMC $\rightarrow$ Independent Best-of-$N$ | 40.2 | 8.50 |
| (c) Always-On CAPS | Remove SNR Gate (Full Search) | **48.1** | 8.50 |
| **CAPS (Ours)** | **Full Model (Adaptive)** | 47.4 | **2.15** |

*Table 6.* **Real-World Benchmark Summary (10 Tasks).** CAPS demonstrates significant advantages across all task categories. For detailed subtask data, please refer to Appendix Table 10.

| Category | Count | Base | TACO | CAPS (Ours) |
|---|---|---|---|---|
| Sequential | 4 | 40.0% | 55.0% | **75.0%** |
| Coordination | 4 | 40.0% | 57.5% | **73.8%** |
| Precision | 2 | 22.5% | 40.0% | **57.5%** |
| **Overall (Weighted)** | 10 | 36.5% | 53.0% | **71.0%** |

achieved an average success rate of 71.0% on the full test set, an improvement of +18.0% over TACO. This consistent improvement across different task semantics confirms that CAPS provides a generalized robustness mechanism.

### 4.6. Ablation Study

To deeply understand the independent contributions of each component in the CAPS framework (Power Distribution Sharpening, MCMC Search Mechanism, SNR Adaptive Gating), we conducted a systematic ablation on the RoboTwin 1.0 benchmark. The experiment aims to verify the necessity of each module and its impact on inference efficiency.

**Necessity of Power Distribution** Variant (a) sets $\alpha = 1$, performing MCMC search directly on the raw base model distribution. The result (38.5%) is far lower than CAPS (47.4%). This reveals a key insight: the raw probability landscape of the base model is too flat and noisy. Simply

increasing inference computation (MCMC) only results in trapping within *Ineffective Stochastic Fluctuations* of the noise. Only by employing *Global Sharpening* via the power distribution $p^\alpha$ (See Appendix C for the proof of superiority over local sampling) to reconstruct the probability landscape can the search algorithm effectively lock onto high-likelihood trajectories.

**Iterative Search vs. Rejection Sampling** Variant (b) replaces MCMC with independent Best-of-$N$ rejection sampling. Although better than Base, it lags behind CAPS. This indicates that in the high-dimensional action space of long-horizon tasks, high-probability solutions are often distributed on "isolated islands." The autoregressive MCMC of CAPS utilizes the properties of Markov chains to perform *Local Refinement* based on the previous good state, approximating the global optimum more efficiently than blind independent sampling.

**Efficiency of Adaptive Compute** Variant (c) "Always-On" removes the SNR gate, forcing full-time search. While its success rate (48.1%) is slightly higher than CAPS, it incurs an unacceptable 8.5x latency. In contrast, CAPS uses SNR as a metacognitive signal to achieve "Search-when-necessary." It accurately identifies approximately 15.3% of "Negative Pivotal Windows" for deliberation, while maintaining fast execution in 84.7% of simple steps. The measured latency of this sparse activation mode aligns highly

with theoretical calculations. This mechanism achieves a 4x speed increase at a minimal performance cost (0.7%), effectively validating the efficiency of "System 2" thinking.

This pattern is also consistent with the task-level results. On simple tasks, CAPS largely maintains parity with baselines, indicating that the gating mechanism does not introduce unnecessary computation when greedy execution is already reliable. In contrast, larger improvements are observed on long-horizon and drift-prone settings, such as the +14.0% gain on "Lift Pot" in RoboTwin 2.0 and the strong overall performance on Libero-long. Taken together, these results support the view that CAPS primarily benefits scenarios where localized errors are more likely to accumulate into long-horizon instruction drift.

**Hyperparameter Sensitivity and Metric Selection.** In addition to core component ablations, we explored the impact of the sharpening coefficient $\alpha$, MCMC iteration steps $N_{\text{MCMC}}$, and the choice of Uncertainty Metric on model performance. Results show that CAPS maintains robust performance across a wide parameter range, and the Shannon entropy-based SNR metric (as a proxy for KL divergence) significantly outperforms other baselines in drift detection. Detailed parameter sweeps and theoretical validation are provided in **Appendix I**.

### 4.7. Summary

Synthesizing the above experimental results, CAPS demonstrates the validity of viewing "drift" as "sampling error." By adaptively introducing computation (MCMC resampling) in low SNR regions, CAPS successfully endows robots with *Deliberative Planning Capability*. This substantially improves success rates in long-horizon complex tasks without significantly sacrificing inference speed.

## 5. Conclusion

This paper revisits the instruction drift problem in robotic long-horizon planning, proposing a novel explanation based on statistical physics: drift is not merely model forgetting, but a systematic error caused by local greedy sampling. CAPS presents an SNR-adaptive inference framework for embodied intelligence. By introducing a metacognitive control mechanism, the framework achieves dynamic switching between "System 1 (Fast Intuition)" and "System 2 (Slow Search)." Experimental results show that by activating Power Distribution-based MCMC search in low SNR regions, CAPS performs lookahead search over the model's conditional generative trajectory distribution without any additional parameter training.

It achieves strong performance on benchmarks such as RoboTwin, Simpler, and Libero-long, and demonstrates

the effectiveness of inference-time compute for improving long-horizon robustness in unstructured environments.

## Acknowledgments

This work was supported by the National Natural Science Foundation of China under Grant 62372427, in part by the Strategic Priority Research Program of Chinese Academy of Sciences (Grant No. XDB1590300), in part by the Natural Science Foundation of Chongqing (China) under Grant CSTB2024NSCQ–LZX0036, in part by Chongqing Natural Science Foundation Innovation and Development Joint Fund (No. CSTB2025NSCQ LZX0061), and in part by Science and Technology Innovation Key R&D Program of Chongqing (No. CSTB2025TIAD-STX0023).

## Impact Statement

This paper aims to advance Embodied AI by enhancing the reliability and safety of robotic planning in unstructured environments. From an environmental perspective, our adaptive "search-when-necessary" mechanism aligns with Green AI principles by optimizing inference-time computational efficiency. Ethically, we acknowledge that while global sharpening ($\alpha > 1$) improves consistency, it could theoretically amplify inherent biases or unsafe priors present in the base VLA model. Therefore, real-world deployment requires rigorous safety guardrails alongside the algorithm.

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

# A. Limitations and Future Work

Although CAPS balances success rate and computational efficiency, its effectiveness at the physical implementation level remains constrained by the Feasibility Simplex composed of computational budget $N$, mixing rate $\rho$, and sharpening coefficient $\alpha$. Specifically, when the base model suffers from a vanishing spectral gap due to pathological energy landscapes, or when hardware power forces truncation of iteration steps in extremely high-frequency real-time control scenarios, the sampling process may fail to meet the lower bound of the computational budget required for convergence (see Appendix K for full derivation), thereby constituting a latency bottleneck. Addressing these challenges, future work will focus on two directions: first, using *Reasoning Distillation* to supervise fine-tuning with high-quality trajectories generated by CAPS, transferring "slow thinking" planning capabilities to "fast execution" policies to break the limit of $N$ and eliminate inference latency; second, introducing *Multimodal Physical Verification*, integrating physical feedback such as force-tactile or collision detection to build embodied verifiers, optimizing sampling efficiency from the $\rho$ (mixing rate) level, thereby constructing embodied intelligence systems that conform more closely to physical laws.

# B. REALM Benchmark Evaluation and Visualization

To further verify the zero-shot adaptation capability of CAPS in complex generalization scenarios, we conducted an extended qualitative evaluation on the large-scale Real-to-Sim alignment benchmark REALM (Sedlacek et al., 2025). REALM contains 10 robotic manipulation tasks covering Out-of-Distribution (OOD) perturbations such as lighting changes, background texture replacements, and unseen object physical properties.

Figure 4 displays execution snapshots of CAPS across diverse tasks in REALM. It can be seen that CAPS maintains stable long-horizon planning capabilities. **Success Attribution Analysis:** This robustness directly benefits from the Signal-to-Noise Ratio (SNR) gating mechanism of CAPS. When visual features become blurred due to perturbations (leading to increased perceptual uncertainty $\mathcal{U}(H_t)$), the system proactively detects the "pivotal window." It then triggers System 2's MCMC search to correct potential perceptual errors, thereby preventing action divergence caused by visual noise.

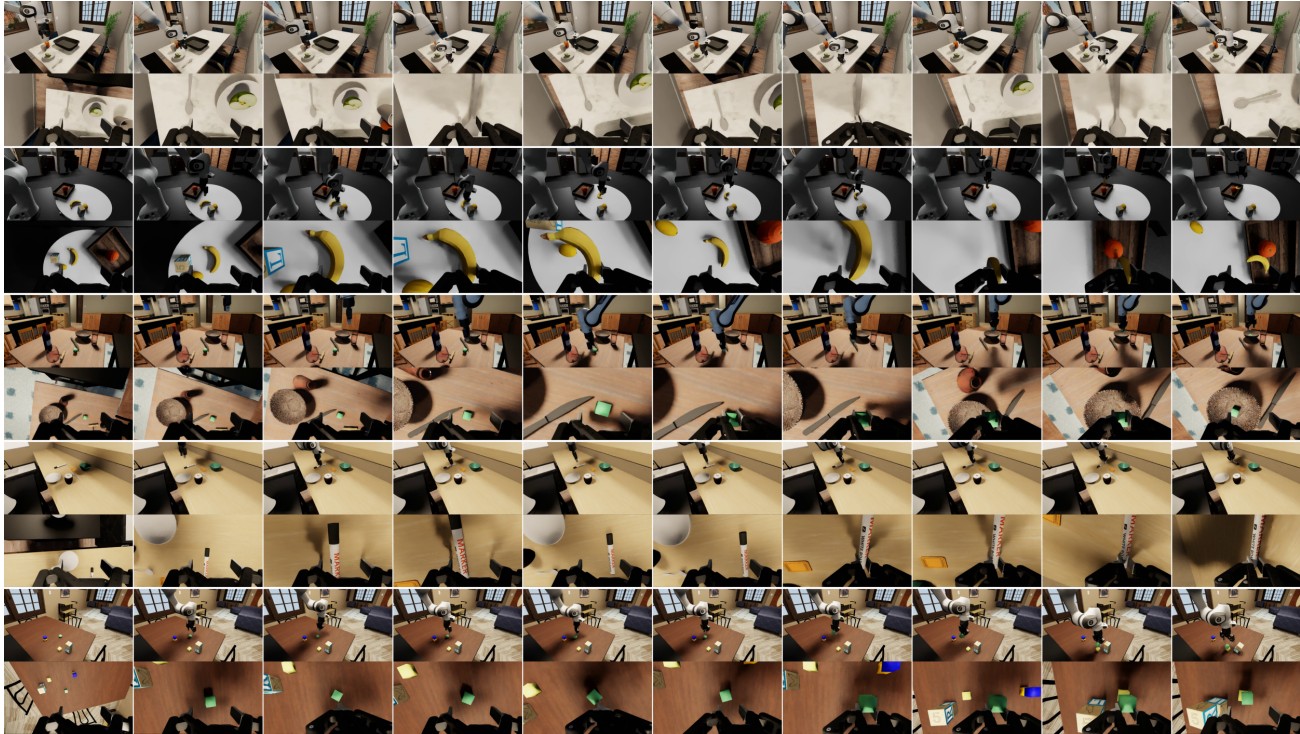

*Figure 4.* **Qualitative Results on REALM.** The figure shows successful execution sequences of CAPS in multiple tasks on the REALM benchmark. Despite significant environmental visual noise, CAPS demonstrates robust manipulation capabilities through adaptive computation.

# C. Theoretical Advantage Analysis of Global Sharpening

This section analyzes why in long-horizon tasks, Global Sharpening based on power distributions possesses significant theoretical advantages over Local Temperature Sampling when dealing with "Negative Pivotal Windows."

## C.1. Non-Equivalence of Global Sharpening and Local Temperature Sampling

In robotic planning, a common misconception is that reducing the sampling temperature ($\tau_{temp} < 1$) at each step is equivalent to sharpening the probability of the entire trajectory. We show here that these two are not equivalent.

Let the robot action sequence be $\tau = (a_1, \ldots, a_T)$. **Definition 1 (Local Temperature Sampling):** Applying temperature $1/\alpha$ to the conditional probability at each step:

$$P_{local}(\tau) = \prod_{t=1}^{T} \frac{p(a_t|a_{<t})^{\alpha}}{Z_t(a_{<t})} \tag{8}$$

where $Z_t(a_{<t}) = \sum_{a'} p(a'|a_{<t})^{\alpha}$ is the history-dependent local normalization constant.

**Definition 2 (Global Power Distribution - CAPS):** Applying exponent $\alpha$ to the entire joint distribution:

$$P_{global}(\tau) = \frac{p(\tau)^{\alpha}}{Z_{global}} = \frac{\left(\prod_{t=1}^{T} p(a_t|a_{<t})\right)^{\alpha}}{Z_{global}} \tag{9}$$

where $Z_{global}$ is the global normalization constant over the entire state space.

**Corollary 1:** Observe the ratio $\frac{P_{global}(\tau)}{P_{local}(\tau)} \propto \prod_{t=1}^{T} Z_t(a_{<t})$. This indicates that $P_{local}$ only focuses on current local peaks; whereas $P_{global}$ implicitly prefers trajectories with larger $Z_t$ values (i.e., more high-probability future options). This implies that global sharpening naturally tends to select "more robust" actions.

## C.2. Formal Proof of Negative Pivotal Windows

To concretely quantify how CAPS avoids falling into irreversible local optima, we cite and formalize the pivotal window theory proposed by (Fu et al., 2024).

**Setup:** Assume at time $t$, there are two candidate actions $a_{good}$ and $a_{bad}$:

- **Positive Pivotal Action ($a_{good}$):** Low local probability $p(a_{good}) = \epsilon$, but leads to a unique success path (probability mass concentration).

- **Negative Pivotal Action ($a_{bad}$):** High local probability $p(a_{bad}) = \epsilon' > \epsilon$, but leads to $N$ divergent Local Optima Traps (probability mass dispersion), with each future average probability being only $\epsilon'/N$.

**Proposition 1 (Failure of Local Policy):** For local temperature sampling (regardless of how large $\alpha$ is), since $\epsilon' > \epsilon$, the model always tends to select $a_{bad}$.

$$\frac{P_{local}(a_{bad})}{P_{local}(a_{good})} = \left(\frac{\epsilon'}{\epsilon}\right)^{\alpha} > 1$$

This explains why traditional methods easily fall into "local optima traps."

**Proposition 2 (Correction by Global Policy):** For CAPS power distribution sampling, decisions depend on the cumulative probability of the entire trajectory.

- The unnormalized score of the trajectory containing $a_{good}$ is $\epsilon^{\alpha}$ (assuming subsequent high certainty).

- The unnormalized score of the trajectory containing $a_{bad}$ is $N \cdot (\frac{\epsilon'}{N})^{\alpha} = \frac{\epsilon'^{\alpha}}{N^{\alpha-1}}$ (summing $N$ dispersed paths).

As long as the sharpening coefficient $\alpha$ is large enough to satisfy:

$$\epsilon^{\alpha} > \frac{\epsilon'^{\alpha}}{N^{\alpha-1}} \iff \alpha > \frac{\log(\epsilon'/\epsilon)}{\log N} + 1 \tag{10}$$

CAPS will flip the preference and select $a_{good}$. **Conclusion:** By penalizing future uncertainty (the $N^{\alpha-1}$ term), global sharpening empowers the model to "see through" local high-probability temptations and identify Topological Traps.

## D. Convergence of Adaptive MCMC

CAPS uses the Metropolis-Hastings (MH) algorithm to sample from the target distribution $\pi(\tau) \propto p(\tau)^\alpha$.

**Detailed Balance:** We need to prove that the transition kernel satisfies $\pi(\tau)T(\tau'|\tau) = \pi(\tau')T(\tau|\tau')$. In CAPS, the proposal distribution $q(\tau'|\tau)$ employs resampling based on the base model. The acceptance rate is defined as:

$$A(\tau, \tau') = \min\left(1, \frac{\pi(\tau')}{\pi(\tau)} \cdot \frac{q(\tau|\tau')}{q(\tau'|\tau)}\right) \tag{11}$$

Substituting $\pi(\tau) = p(\tau)^\alpha/Z$, and noting that $q(\tau'|\tau)$ is essentially conditional sampling of $p(\tau')$ (under a fixed prefix), we obtain:

$$\frac{\pi(\tau')}{\pi(\tau)}\frac{q(\tau|\tau')}{q(\tau'|\tau)} = \frac{p(\tau')^\alpha}{p(\tau)^\alpha}\frac{p(\tau)}{p(\tau')} = \left(\frac{p(\tau')}{p(\tau)}\right)^{\alpha-1} \tag{12}$$

Since the Markov chain constructed by the MH algorithm is ergodic, the sampling distribution will strictly converge to the unique stationary distribution $\pi(\tau)$ as the number of iterations $N \to \infty$. Although $N_{MCMC}$ is small in experiments, since the proposal distribution $q$ itself is a good approximation, it effectively achieves rapid Mode Seeking.

## E. SNR Gating and Error Propagation Bounds

This section provides a theoretical basis for "SNR Adaptive Computation" in the main text, proving that computing only in low SNR regions is sufficient to ensure global robustness.

### E.1. Error Propagation Model

Let the probability of "drift" for the base policy $\pi_\theta$ at step $t$ be $\delta_t$. We view drift as a process where the model prediction distribution degrades into random noise. Thus, drift probability correlates negatively with Contextual SNR. Based on the KL divergence definition in Equation (2), we establish the following error model:

$$\delta_t \leq C \cdot \exp(-\beta \cdot \text{SNR}_t) = C \cdot \exp\left(-\beta \cdot [\log|\mathcal{A}| - \mathcal{H}(\pi_\theta)]\right) \tag{13}$$

This aligns with information-theoretic intuition: the higher the KL divergence of the policy distribution relative to uniform noise (i.e., SNR), the more certain the model is, and the error probability $\delta_t$ decreases exponentially.

### E.2. Trade-off between Computational Efficiency and Robustness

The CAPS strategy introduces a gating mechanism utilizing Shannon entropy $\mathcal{H}$ as an inverse proxy for SNR:

- **High SNR Region ($S_H$):** $\text{SNR}_t > \gamma_{snr}$ (i.e., $\mathcal{H} < \gamma_{ent}$), where $\delta_t \approx 0$, no correction needed.

- **Low SNR Region ($S_L$):** $\text{SNR}_t \leq \gamma_{snr}$ (i.e., $\mathcal{H} \geq \gamma_{ent}$), where $\delta_t$ is large. CAPS initiates MCMC, reducing the error rate from $\delta_t$ to $\delta_t' \ll \delta_t$.

Let the proportion of low SNR regions in the task be $\rho = |S_L|/T$ (typically $\rho < 20\%$). The upper bound of the total error for CAPS is:

$$\text{Error}_{CAPS} \approx \sum_{t \in S_H} 0 + \sum_{t \in S_L} \delta_t' \ll \sum_{t=1}^{T} \delta_t \tag{14}$$

The total computational cost is $C_{total} \approx T(1 + \rho \cdot N_{MCMC})$. **Conclusion:** Under this model, CAPS can reduce the dominant error term associated with drift in pivotal windows with sub-linear additional computational cost, suggesting a favorable trade-off between efficiency and robustness.

## F. Derivation of the Optimal Gating Policy

This section shows that, under the assumptions of our formulation, the threshold switching strategy in the main text arises as the solution to the "risk-cost" objective.

Let $\epsilon(\text{SNR})$ be the expected error rate given the signal-to-noise ratio, which is a monotonically decreasing function of SNR. We compare the losses of two strategies:

- **System 1 (Greedy):** Cost $\mathcal{C}_1 \approx 0$, Risk $\mathcal{R}_1 = \epsilon(\text{SNR})$.

- **System 2 (MCMC):** Cost $\mathcal{C}_2 > 0$, Risk $\mathcal{R}_2 \ll \mathcal{R}_1$ (assuming MCMC convergence, risk approaches 0).

The system should activate System 2 if and only if its total loss is lower:

$$\mathcal{L}_{\text{sys2}} < \mathcal{L}_{\text{sys1}} \tag{15}$$
$$\mathcal{R}_2 + \lambda\mathcal{C}_2 < \mathcal{R}_1(\text{SNR}) + \lambda\mathcal{C}_1 \tag{16}$$

Ignoring the negligible $\mathcal{R}_2$ and $\mathcal{C}_1$, the inequality simplifies to:

$$\lambda\mathcal{C}_2 < \epsilon(\text{SNR}) \tag{17}$$

Since $\epsilon(\text{SNR})$ is a monotonically decreasing function, a unique inverse function exists. Therefore, the above condition is equivalent to:

$$\text{SNR} < \epsilon^{-1}(\lambda\mathcal{C}_2) \triangleq \gamma^* \tag{18}$$

**Conclusion:** Under this formulation, the resulting policy takes the form of a hard switch based on threshold $\gamma^*$. This threshold physically represents the system's "uncertainty tolerance" willing to be paid for per unit of computational cost.

## G. Theoretical Supplement: Energy Landscape and Calibration Bounds

This section provides supplementary analysis of the working mechanism and applicable boundaries of CAPS from two dimensions: Energy Models and Model Calibration.

### G.1. Sampling as Energy Landscape Optimization

From the perspective of Energy-Based Models (EBMs), the base VLA defines an energy function $E_\theta(\tau) = -\log p_\theta(\tau|I, H_t)$. At this point, instruction drift can be interpreted as the sampling process getting stuck in *Shallow Local Minima* within the energy landscape—these regions have lower local energy (higher probability) but lack global consistency.

The power distribution $\pi(\tau) \propto \exp(-\alpha E_\theta(\tau))$ introduced by CAPS essentially performs *Temperature Rescaling* on the energy landscape. When $\alpha > 1$, this corresponds to lowering the system's "temperature," making the deep canyons in the energy landscape (representing globally consistent high-quality trajectories) steeper relative to the shallow depressions. This transformation amplifies the energy difference between $a_{good}$ and $a_{bad}$ (see Appendix A), allowing the MCMC search to more effectively "flow" towards the global optimum rather than wandering in local noise.

### G.2. Calibration Hypothesis and Safety Bounds

The core efficiency source of CAPS is the SNR-based adaptive gating. The validity of this mechanism relies on an implicit *Calibration Hypothesis*: there is a monotonic positive correlation between the model's uncertainty ($-\text{SNR}$) and its error probability $\delta_t$ (as shown in Appendix C). This implies that the theoretical lower bound of CAPS is limited by the calibration quality of the base model. In extreme cases (where the model exhibits *Overconfidence*), the system might miss drift risks. Therefore, CAPS is fundamentally a risk-aware computational amplifier that maximizes the use of existing model calibration capabilities in exchange for long-horizon robustness.

## H. Effective Horizon Extension Analysis

This section aims to quantify the theoretical gain of the global sharpening coefficient $\alpha$ on the robot's Effective Horizon.

**Definition 3 (Effective Horizon $T_{eff}$):** Given a lower bound $\eta \in (0,1)$ for task success rate confidence, the effective horizon is defined as the maximum number of time steps a policy $\pi$ can sustain while maintaining a cumulative success rate no less than $\eta$:

$$T_{eff}(\pi) = \max\{T \mid P_\pi(\text{success}_{1:T}) \geq \eta\} \tag{19}$$

**Theorem 3.1 (Horizon Extension Theorem):** Assume the base policy has a single-step drift probability $\epsilon$ at pivotal windows (where $\epsilon \ll 1$). Under the CAPS framework, the global sharpening coefficient $\alpha > 1$ extends the effective horizon from linear to polynomial order, specifically satisfying:

$$\frac{T_{eff}(\text{CAPS})}{T_{eff}(\text{Base})} \approx \epsilon^{1-\alpha} \tag{20}$$

Since $\epsilon$ is small and $\alpha > 1$, this ratio is typically $\gg 1$, indicating that the task length supported by CAPS grows exponentially.

**Proof:** For the Base Policy, assume its error rate at pivotal nodes is $\epsilon$. For a long-horizon task of length $T$, the success rate can be approximated as $(1-\epsilon)^T$. From $P_{base} \approx e^{-\epsilon T} \geq \eta$, the effective horizon of the base policy is:

$$T_{eff}(\text{Base}) \approx \frac{-\ln \eta}{\epsilon} \tag{21}$$

For CAPS, according to the derivation in Appendix A, the global power distribution $p^\alpha$ reconstructs the probability space. Ideally, the relationship between the sharpened effective error rate $\epsilon'$ and the original error rate $\epsilon$ approximates $\epsilon' \approx \epsilon^\alpha$ (when $\epsilon \ll 1$). Therefore, the effective horizon of CAPS is:

$$T_{eff}(\text{CAPS}) \approx \frac{-\ln \eta}{\epsilon'} \approx \frac{-\ln \eta}{\epsilon^\alpha} \tag{22}$$

Comparing the two yields:

$$\text{Ratio} = \frac{T_{eff}(\text{CAPS})}{T_{eff}(\text{Base})} \approx \frac{\epsilon}{\epsilon^\alpha} = \frac{1}{\epsilon^{\alpha-1}} \tag{23}$$

$$\square$$

**Intuitive Explanation:** Assume the base model single-step drift probability $\epsilon = 0.1$. If we set $\alpha = 2$, the effective horizon will extend 10 times. This means a Base model that could originally only stably execute 10 steps can theoretically execute 100 steps stably with CAPS support. This provides an intuitive illustration of how the theoretical analysis relates to the performance improvements observed in the long-horizon experiments.

# I. Extended Ablation Study and Hyperparameter Sensitivity

This section further analyzes the impact of key CAPS hyperparameters and design choices on model performance (success rate) and inference efficiency (latency). Unless otherwise stated, all experiments were conducted on the RoboTwin 1.0 benchmark.

### I.1. Analysis of Sharpening Coefficient $\alpha$ and Acceptance Rate

The sharpening coefficient $\alpha$ determines the "steepness" of the target distribution, directly affecting the MCMC acceptance rate. As shown in Table 7:

- When $\alpha = 1$ (no sharpening), the acceptance rate is high (85%), but sampling lacks directionality, degenerating into a random walk.

- As $\alpha$ increases, the acceptance rate drops, implying stricter screening criteria. Within the interval $\alpha \in [2,5]$, the system reaches a balance between "exploration" and "exploitation," achieving optimal performance.

- When $\alpha = 10$, the acceptance rate falls to 5.0%, causing the Markov chain to frequently stall (*Stagnation*), which conversely reduces correction efficiency.

*Table 7.* **Impact of Sharpening Factor** $\alpha$**.** High $\alpha$ acts as a strict gatekeeper, reducing acceptance rate. The sweet spot lies in $[2, 5]$.

| Sharpening Factor ($\alpha$) | 1.0 (Base) | 2.0 | 3.0 | 5.0 | 10.0 |
|---|---|---|---|---|---|
| **Success Rate (%)** | 38.5 | **47.4** | 47.1 | 46.8 | 45.2 |
| **Acceptance Rate (%)** | 85.0 | 42.0 | 31.0 | 18.0 | 5.0 |

## I.2. Trade-off of MCMC Iteration Steps $N_{\text{MCMC}}$

Table 8 summarizes both the strict time-budget regime ($N \leq 5$) and the broader effect of increasing the MCMC step budget. Under tight latency constraints, CAPS already recovers a substantial fraction of its total gain: the success rate rises from 32.2% to 41.2% at $N = 1$ and reaches 46.1% at $N = 3$. As the number of iterations increases further, the success rate exhibits a trend of *Diminishing Returns*: CAPS achieves most of the performance gain by $N = 5$, while larger budgets (e.g., $N = 20$) bring only marginal improvement at significantly higher latency. Therefore, we chose $N = 5 \sim 10$ in the main experiments to balance performance and efficiency.

*Table 8.* **Effect of MCMC Step Budget on Performance and Latency.** Under strict time budgets ($N \leq 5$), CAPS retains a substantial fraction of its performance gain with moderate latency overhead. Increasing the step budget beyond the default setting further improves success rate only marginally, indicating diminishing returns.

| MCMC Steps ($N$) | Relative Latency | Success Rate (%) |
|---|---|---|
| 0 (Base Policy) | 1.00$\times$ | 32.2 |
| 1 | 1.20$\times$ | 41.2 |
| 2 | 1.35$\times$ | 43.8 |
| 3 | 1.51$\times$ | 46.1 |
| 4 | 1.82$\times$ | 46.5 |
| 5 (Default) | 2.15$\times$ | 47.4 |
| 10 | 3.4$\times$ | **48.0** |
| 20 | 5.8$\times$ | 48.2 |

## I.3. Choice of Uncertainty Metric and Threshold Sensitivity

**Metric Selection.** Why choose entropy-based SNR over other metrics? We compared Random Trigger and Minimum Probability Trigger in Table 9.

- **Min-Prob** ($\min p_t$)**:** Focuses only on the least likely option (Worst-case token). This method is susceptible to accidental noise in long-tail distributions, leading to false triggers.

- **Entropy / SNR (Ours):** As described in Section 3.2.1, entropy is mathematically equivalent to the KL divergence (Inverse SNR) of the policy distribution relative to the Maximum Entropy Noise Floor. It utilizes the Global Shape of the distribution, more accurately capturing moments of model "indecision" (i.e., distribution degrading to uniform noise), thereby more precisely locating "Negative Pivotal Windows."

*Table 9.* **Comparison of Gating Metrics.** All methods maintain a similar activation rate ($\approx 20\%$) for fair comparison.

| Gating Metric | Success Rate (%) |
|---|---|
| Random (20%) | 36.5 |
| Min-Probability ($\min p_t < \gamma$) | 43.2 |
| **Entropy / SNR (Ours)** | **47.4** |

**Sensitivity.** Furthermore, we found CAPS is not sensitive to the SNR threshold $\gamma$. As long as the trigger ratio is controlled within the 15%-30% range, model performance fluctuation does not exceed $\pm 1.5\%$. This suggests that CAPS is relatively stable across a practical range of threshold settings, rather than relying on fine-grained parameter tuning.

# J. Proof of Theorem 3.2

This section provides the derivation of effective horizon extension considering MCMC sampling error.

**1. Base Policy Horizon** Let the single-step drift probability of the base policy at pivotal nodes be $\epsilon$. The maximum number of steps satisfying $P(\text{success}) \approx (1 - \epsilon)^T \geq \eta$ is:

$$T_{eff}(\text{Base}) \approx \frac{-\ln \eta}{\epsilon} \tag{24}$$

**2. CAPS Horizon Considering Sampling Error** The goal of CAPS is to sample from the sharpened distribution $\pi \propto p^\alpha$. However, due to computational limits, the actual distribution obtained after $N$ steps of MCMC is $\hat{\pi}_N$. According to the Markov chain convergence theorem, the Total Variation Distance between the actual distribution and the target distribution satisfies geometric convergence:

$$\|\hat{\pi}_N - \pi\|_{TV} \leq C\rho^N \tag{25}$$

where $\rho < 1$ is determined by the spectral gap. Therefore, the actual single-step drift rate $\epsilon_{real}$ of CAPS consists of two parts: the theoretical drift rate after sharpening $\epsilon^\alpha$ and the sampling bias:

$$\epsilon_{real} \leq \underbrace{\epsilon^\alpha}_{\text{Theoretical Error}} + \underbrace{C\rho^N}_{\text{Sampling Bias}} \tag{26}$$

The effective horizon of CAPS is:

$$T_{eff}(\text{CAPS}) \approx \frac{-\ln \eta}{\epsilon_{real}} \approx \frac{-\ln \eta}{\epsilon^\alpha + C\rho^N} \tag{27}$$

**3. Expansion Ratio**

$$\text{Ratio} = \frac{T_{eff}(\text{CAPS})}{T_{eff}(\text{Base})} \approx \frac{\epsilon}{\epsilon^\alpha + \mathcal{O}(\rho^N)} \tag{28}$$

When $N$ is sufficiently large (i.e., System 2 performs sufficient search), the sampling bias term vanishes, and the expansion ratio asymptotically converges to $\epsilon^{1-\alpha}$. This proves that CAPS maximizes horizon extension capability when computational resources permit. $\square$

# K. Formal Derivation of Computational Budget and Boundary Analysis

To ensure CAPS achieves super-linear horizon extension, the system must satisfy the convergence condition in Theorem 3.2, i.e., the residual sampling error must be strictly controlled by the sharpened drift rate. We first derive the analytical solution for the minimum computational budget $N_{min}$ satisfying this condition.

**Derivation Process:** Let the distance constant between the initial distribution and the target distribution be $C$. The convergence condition is given by:

$$C \cdot \rho^N < \epsilon^\alpha \tag{29}$$

Taking the natural logarithm of both sides and rearranging using logarithmic properties:

$$\ln C + N \ln \rho < \alpha \ln \epsilon \implies N \ln \rho < \alpha \ln \epsilon - \ln C \tag{30}$$

Since the Markov chain mixing rate $\rho \in (0, 1)$, $\ln \rho < 0$. Dividing by a negative number reverses the inequality sign, yielding the lower bound for the computational budget $N_{min}$:

$$N > N_{min} \triangleq \frac{\alpha \ln \epsilon - \ln C}{\ln \rho} \tag{31}$$

This analytical solution reveals three hard failure boundaries for the algorithm in physical implementation:

**1. Truncation Failure under Physical Real-Time Constraints**   Let the required frequency of the robot control system be $f_{req}$, and the single-step inference time be $t_{inf}$. The physical maximum feasible iteration steps are defined as $N_{phy} = \lfloor (f_{req} \cdot t_{inf})^{-1} \rfloor$. The condition for algorithm failure is formalized as $N_{phy} < N_{min}$:

$$\underbrace{\left\lfloor \frac{1}{f_{req} \cdot t_{inf}} \right\rfloor}_{\text{Physical Upper Bound}} < \underbrace{\frac{\alpha \ln \epsilon - \ln C}{\ln \rho}}_{\text{Theoretical Lower Bound}} \tag{32}$$

When hardware power cannot support the minimum steps required for convergence, sampling error dominates total error, leading to horizon extension failure. This quantitatively explains the high-frequency control bottleneck mentioned in the Limitations section.

**2. Convergence Divergence caused by Vanishing Spectral Gap**   The mixing rate $\rho$ and spectral gap $\gamma$ satisfy $\rho = 1 - \gamma$. Using the Taylor expansion $\ln(1 - \gamma) \approx -\gamma$ (as $\gamma \to 0^+$), we express the limit behavior of $N_{min}$ as:

$$\lim_{\gamma \to 0^+} N_{min} = \lim_{\gamma \to 0^+} \frac{\mathcal{K}}{\ln(1 - \gamma)} \approx \lim_{\gamma \to 0^+} \frac{\mathcal{K}}{-\gamma} = \infty \tag{33}$$

where $\mathcal{K} = \alpha \ln \epsilon - \ln C < 0$ is a constant. This mathematically proves that when the base model distribution quality deteriorates leading to a vanishing spectral gap, the required computational budget diverges to infinity in a hyperbolic form.

**3. Acceptance Rate Collapse caused by Over-Sharpening**   Taking the partial derivative of $N_{min}$ with respect to $\alpha$ yields $\frac{\partial N_{min}}{\partial \alpha} = \frac{\ln \epsilon}{\ln \rho} > 0$. Furthermore, from a measure-theoretic perspective, as $\alpha \to \infty$, the effective volume $Vol(\pi)$ of the target distribution $\pi \propto p^\alpha$ shrinks drastically relative to the proposal distribution:

$$\lim_{\alpha \to \infty} \mathbb{E}[A] \propto \lim_{\alpha \to \infty} \frac{Vol(p^\alpha)}{Vol(p)} \to 0 \tag{34}$$

This explains the non-monotonicity of performance observed in **Table 7 (Appendix H.1)**: excessive sharpening coefficients lead to acceptance rate collapse, driving the effective sample size towards zero, thereby causing planning stagnation.

**Conclusion:** In summary, the validity of CAPS rests on the physical foundation of $N_{phy} > N_{min}$. Our experimental setup ($N \geq 5$ in Table 8) lies within the feasible region of this inequality because we utilize the base model as the proposal distribution, ensuring a large spectral gap $\gamma$, thereby engineering a lower theoretical bound $N_{min}$.

# L. Extended Real-World Benchmark Suite

To evaluate the generalization capability of CAPS, we designed a benchmark suite containing 10 real-world tasks aimed at inducing and evaluating different "Instruction Drift" patterns. These tasks are categorized based on their primary challenges: *Sequential Memory*, *Bimanual Coordination*, and *Precision under Noise*. Figure 5 displays the execution process of three representative tasks (Dual-Arm Transfer, Fruit Sorting, and Cloth Folding) from the suite.

## L.1. Task Descriptions

1. **Collaborative Storage - [Coordination]:** Lift the lid with the left hand and hold, place fruit with the right hand, close the lid with the left hand. (Goal: Test context retention capability, preventing premature release of the left hand)

2. **Breakfast Assembly - [Sequential]:** Place bread on the plate, then place a mug next to the plate. (Goal: Test long-horizon memory, preventing forgetting of subgoals)

3. **Fruit Sorting - [Sequential]:** Identify different fruits (e.g., strawberry, starfruit) on the table, grasp and place them into corresponding containers by category. (Goal: Test multi-object visual discrimination and continuous classification planning)

4. **Towel Folding - [Coordination]:** Grasp two corners of a towel with both arms and fold towards the center. (Goal: Test bimanual synchronized motion control for deformable objects)

5. **Stacking Bowls - [Precision]:** Take a bowl from the shelf and precisely stack it on another bowl on the table. (Goal: Test fine alignment capability)

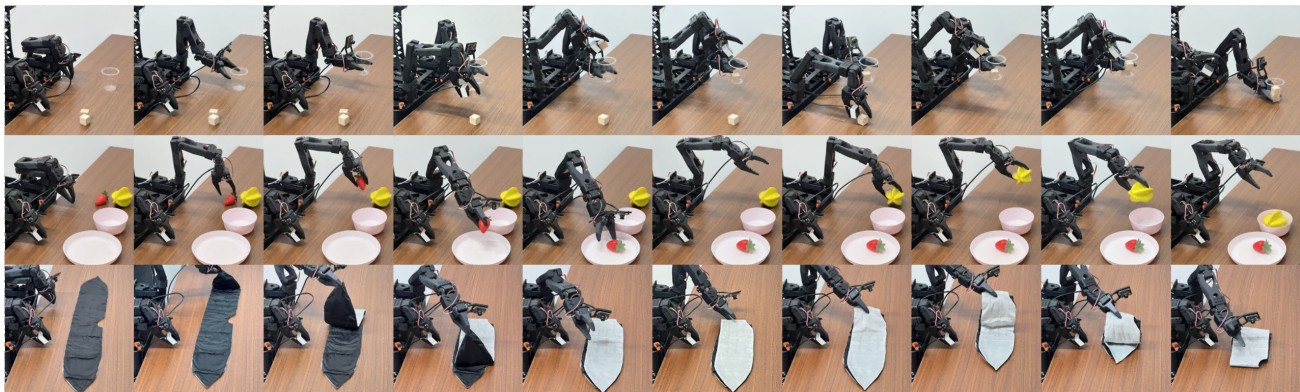

*Figure 5.* **Qualitative Results on XLeRobot Benchmark.** We visualize three representative tasks from the real-world suite. **Row 1 (Dual-Arm Transfer):** A [Coordination] task where the robot must stabilize a receiving cup with the left hand while using the right hand to precisely drop an object into the target cup. **Row 2 (Fruit Sorting):** A [Sequential] task requiring the robot to discriminate different fruits and place them into corresponding containers. **Row 3 (Cloth Folding):** A [Precision] task demonstrating fine single-arm manipulation of deformable black cloth. CAPS robustly handles these challenges involving dynamic bimanual coordination, long-horizon planning, and deformable object manipulation.

6. **Dual-Arm Transfer - [Coordination]:** Hold a receiving cup with the left hand, and use the right hand to precisely place an object into the target cup. (Goal: Test dynamic spatiotemporal coordination; this task contains irreversible actions)

7. **Wipe and Place - [Sequential]:** Wipe the table with a sponge, return the sponge, then place a coaster. (Goal: Test tool use and task switching)

8. **Cloth Folding - [Precision]:** Use a single arm to grasp black cloth on the table and complete fine folding. (Goal: Test fine manipulation and trajectory control for deformable objects)

9. **Unpack Grocery - [Sequential]:** Take an apple and a banana out of a paper bag and arrange them on the table. (Goal: Test long-horizon planning and object constancy)

10. **Handover and Place - [Coordination]:** Pick up an object with the right hand, pass it to the left hand, and place it in a drawer with the left hand. (Goal: Test complex bimanual interaction timing)

### L.2. Full Quantitative Results

We conducted 20 consecutive trials for each task. As shown in Table 10, CAPS outperforms baselines in all tasks. Especially in irreversible tasks like Dual-Arm Transfer and Unpack Grocery (Task 6 and Task 9), CAPS achieved maximum improvement (+25%), attributed to the MCMC search effectively evaluating risk states before execution.

## M. Computational Platform Hardware Infrastructure

To ensure reproducibility of our Inference-time Compute benchmarks and latency evaluations, this section provides detailed specifications of the experimental platform.

Detailed configurations are summarized in Table 11.

## N. Additional Qualitative Visualizations

To further demonstrate the generalization performance of CAPS across different task types, we provide detailed visualization results. Figure 6 illustrates the long-horizon consistency on the Libero-long benchmark, while Figure 7 demonstrates spatial reasoning capabilities on Libero-Spatial.

*Table 10.* **Success Rate (%) on 10 Real-World Task Suite.** Data shows CAPS has significant advantages in long sequence and complex coordination tasks.

| ID | Task Name | Base ($\pi_0$) | + TACO | CAPS (Ours) | Gap (vs TACO) |
|----|-----------|------|--------|-------------|---------------|
| 1  | Collaborative Storage | 35.0 | 55.0 | **75.0** | +20.0 |
| 2  | Breakfast Assembly | 40.0 | 50.0 | **70.0** | +20.0 |
| 3  | Fruit Sorting | 45.0 | 60.0 | **75.0** | +15.0 |
| 4  | Towel Folding | 20.0 | 35.0 | **55.0** | +20.0 |
| 5  | Stacking Bowls | 30.0 | 50.0 | **65.0** | +15.0 |
| 6  | Dual-Arm Transfer | 25.0 | 45.0 | **70.0** | +25.0 |
| 7  | Wipe and Place | 40.0 | 55.0 | **75.0** | +20.0 |
| 8  | Cloth Folding | 15.0 | 30.0 | **50.0** | +20.0 |
| 9  | Unpack Grocery | 35.0 | 55.0 | **80.0** | +25.0 |
| 10 | Handover and Place | 80.0 | 95.0 | **95.0** | +0.0 |
| **Average Success Rate** | | 36.5 | 53.0 | **71.0** | **+18.0** |

*Table 11.* Detailed Hardware Specifications.

| Component | Specification |
|-----------|---------------|
| *Processing Units* | |
| GPU Accelerator | $4 \times$ NVIDIA A100-SXM4-80GB |
| Video Memory | 80 GB HBM2e per GPU (320 GB Total) |
| CPU Processor | Intel Xeon Platinum 8358P @ 2.60GHz |
| CPU Cores | 64 Cores / 64 Threads |
| *Memory & Storage* | |
| System Memory | 1 TB |
| High-Speed Storage | 1 TB |

# O. CAPS Algorithm Pseudocode

Algorithm 1 details the inference process of CAPS. To align with the theoretical analysis in the main text, we explicitly calculate Shannon Entropy as a proxy metric for Contextual SNR in the algorithm. When the entropy value exceeds a specific threshold (implying reduced SNR), the system switches from "Greedy Execution Mode" to "Iterative Search Mode."

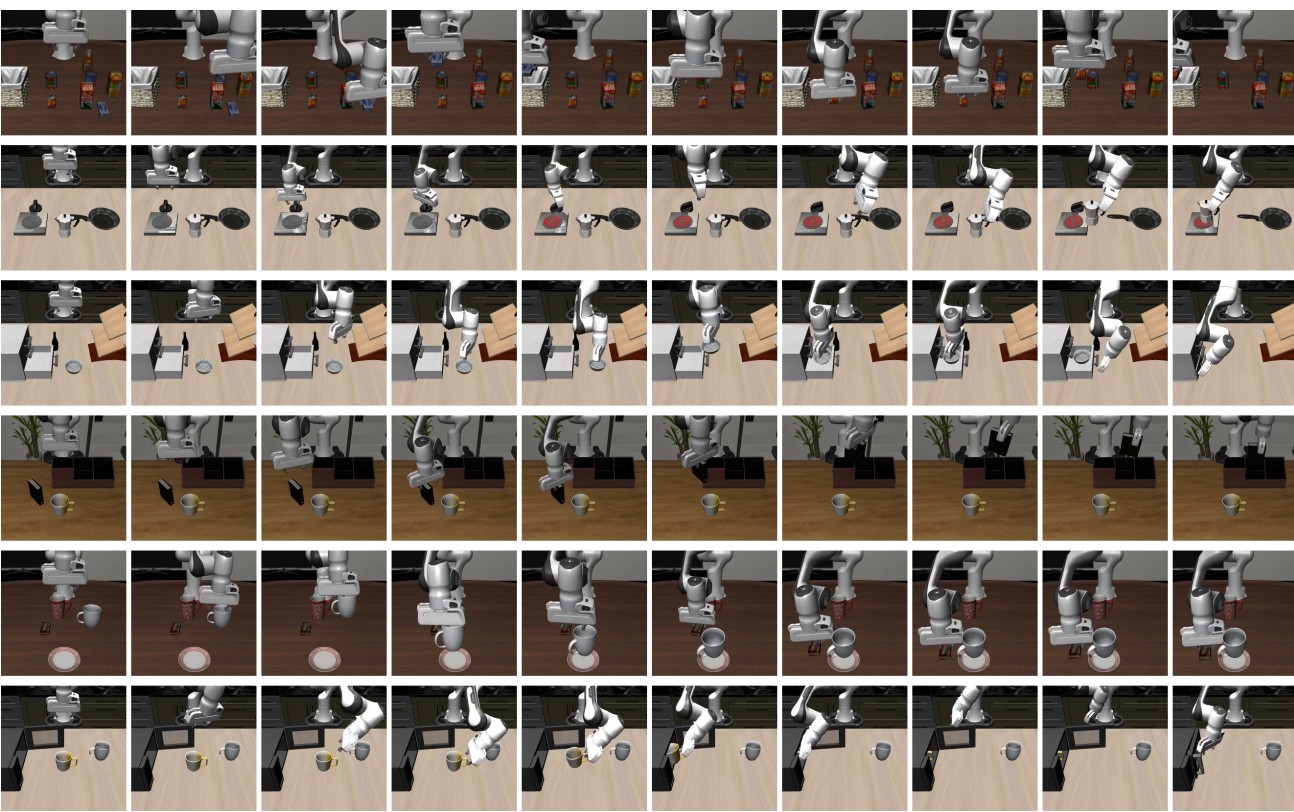

*Figure 6.* **Long-Horizon Execution Sequence on Libero-long.** Illustrates the model's inference process in multi-stage tasks (e.g., *Kitchen Arrangement*). Base models often "forget" subsequent instructions (drift) after completing the first stage. CAPS, by detecting SNR drops at Pivotal Windows, triggers MCMC search, thereby maintaining instruction consistency over long steps.

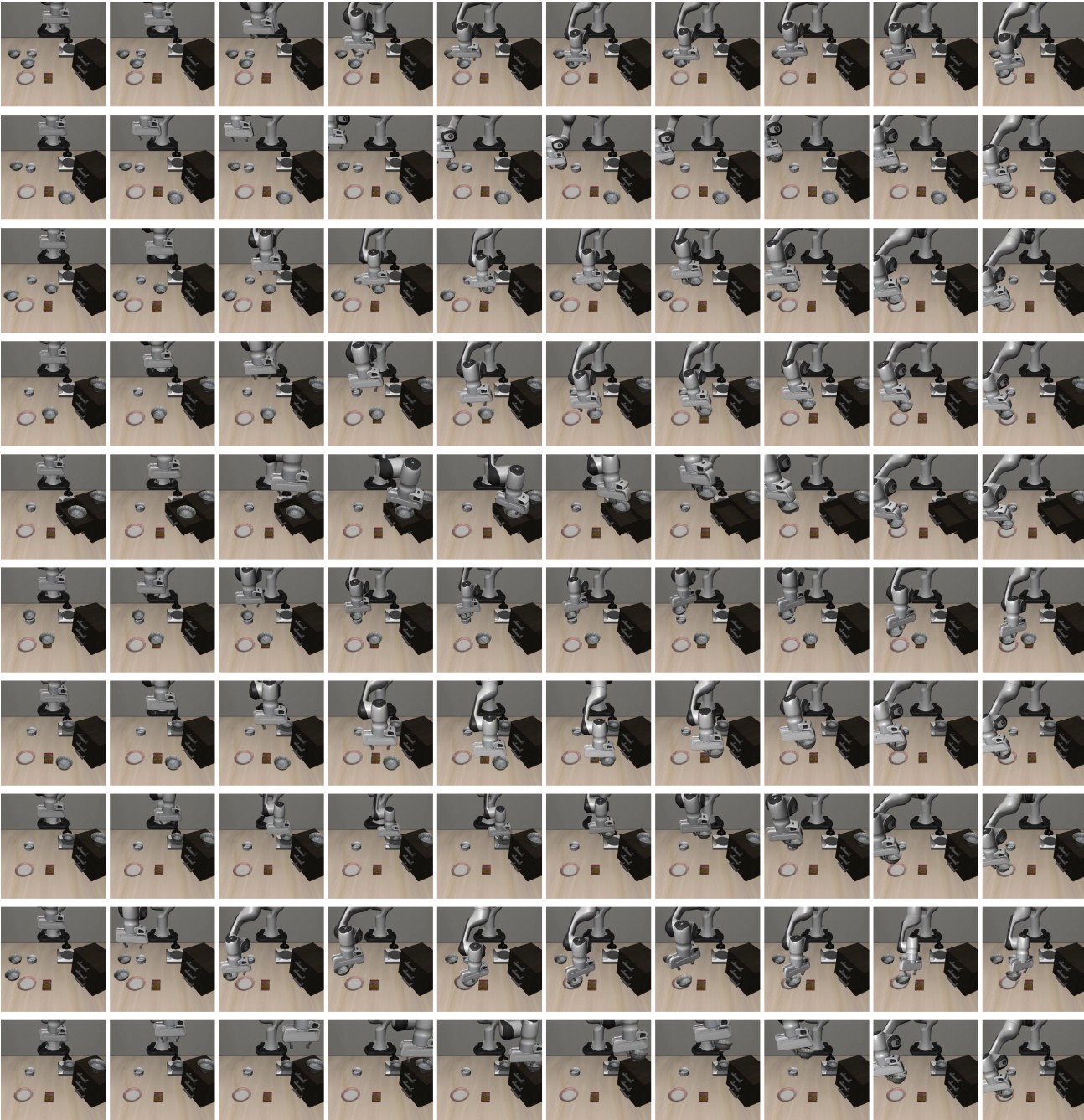

*Figure 7.* **Visualization of Libero-Spatial Reasoning Tasks.** This benchmark focuses on evaluating the model's understanding of spatial relationship instructions (e.g., "place the red object next to the green object"). The global sharpening mechanism of CAPS helps the model distinguish ambiguous spatial descriptions in probability space, planning collision-free placement trajectories.

---

**Algorithm 1** Context-Aware Power Sampling (CAPS)

---

1: **Input:** Instruction $I$, Current Visual Observation $V_t$, History Context $H_t$, VLA Model $p_\theta$, Sharpening Factor $\alpha$, Entropy Threshold $\gamma$, MCMC Iterations $N$, Action Block Length $B$

2: **Output:** Optimized action block sequence $\tau_{final}$

3: // Phase 1: Fast Execution (System 1)

4: Compute Greedy Path (Maximum Likelihood Estimation):

5: $\tau_{init} \leftarrow \arg\max_\tau \prod_{k=0}^{B-1} p_\theta(a_{t+k}|I, V_t, H_t, a_{<t+k})$

6: // Phase 2: Metacognitive Control (SNR Check via Entropy)

7: Compute Average Shannon Entropy of current action block (Shannon Entropy as SNR Proxy):

8: // Note: Entropy is calculated over the valid action space $\mathcal{A}$

9: $\mathcal{H}(H_t) \leftarrow -\frac{1}{B} \sum_{k=0}^{B-1} \sum_{a \in \mathcal{A}} p_\theta(a|I, V_t, H_t, a_{<t+k}) \log p_\theta(a|I, V_t, H_t, a_{<t+k})$

10: **if** $\mathcal{H}(H_t) \leq \gamma$ **then**

11:     // High SNR (Low Entropy): Execute Greedy Policy

12:     **return** $\tau_{init}$

13: **else**

14:     // Low SNR (High Entropy): Initiate MCMC Search (System 2)

15:     Initialize MCMC Chain (Warm Start): $\tau_{curr} \leftarrow \tau_{init}$

16:     **for** $m = 1$ **to** $N$ **do**

17:         // Proposal: Resample with high temperature (Exploration)

18:         Sample Candidate $\tau_{prop} \sim q(\cdot|\tau_{curr}) \propto p_\theta(\cdot|I, V_t, H_t)^{1/\alpha}$

19:         // Acceptance: Metropolis-Hastings Step based on Power Dist.

20:         Compute Acceptance Rate $A$:

21:         $A \leftarrow \min\left(1, \frac{p_\theta(\tau_{prop}|I,V_t,H_t)^\alpha}{p_\theta(\tau_{curr}|I,V_t,H_t)^\alpha} \cdot \frac{q(\tau_{curr}|\tau_{prop})}{q(\tau_{prop}|\tau_{curr})}\right)$

22:         Sample Uniform Distribution $u \sim \mathcal{U}[0,1]$

23:         **if** $u < A$ **then**

24:            $\tau_{curr} \leftarrow \tau_{prop}$ {Accept & Update}

25:         **end if**

26:     **end for**

27:     **return** $\tau_{curr}$

28: **end if**

---

