# OpenReview forum: "Rethinking Instruction Drift as a Sampling Error: SNR-Aware Power Distributions for Long-Horizon Robotic Planning"
_ICML.cc/2026/Conference — ICML 2026 regular_

### Official Review · Reviewer_Hxgt · 2026-03-04

**Soundness:** 2
**Presentation:** 2
**Significance:** 2
**Originality:** 2
**Overall Recommendation:** 4
**Confidence:** 4

**Summary:**

This paper reconceptualizes "instruction drift" in long-horizon robotic Vision-Language-Action (VLA) models not as a memory failure, but as a systematic sampling error where local greedy sampling gets trapped in "Negative Pivotal Windows". To resolve this, the authors introduce Context-Aware Power Sampling (CAPS), a training-free inference-time computation framework. CAPS uses a metacognitive gating mechanism based on Signal-to-Noise Ratio (SNR) to dynamically switch between fast greedy execution and a slow, Iterative Refinement process via Markov Chain Monte Carlo (MCMC). This slow search utilizes a global power distribution to sharpen probabilities, aiming to enforce long-horizon physical consistency.

**Compliance With Llm Reviewing Policy:**

Affirmed.

**Final Justification:**

I am now positive on the paper overall because I find the core method original and practically significant: reframing drift as a sampling problem and combining SNR-gated inference-time refinement with power-distribution sampling is a novel idea, and the experiments show meaningful gains across multiple long-horizon benchmarks and real-world tasks. My remaining reservation is mainly about clarity and claim calibration rather than the usefulness of the method itself: Eq. (1) and Theorems 3.1/3.2 are written at the trajectory/effective-horizon level, while the implemented algorithm is block-based with an explicit action block length B, so the original draft overstated the extent to which CAPS should be read as exact long-horizon global planning. The rebuttal did not fully close this theory-implementation gap, but it did address my main concern by explicitly acknowledging chunk-level refinement as a tractable approximation to the global objective and by committing to revise the paper’s writing and claims accordingly. That changed my evaluation: I now view the main weakness as positioning and exposition, not method validity, and with clearer framing in the final version I believe this will be a valuable contribution.

**Key Questions For Authors:**

- The ablation study shows a 2.15x relative latency for CAPS. When the robot dynamically switches to "System 2" during a fluid physical task, does this sudden computational delay cause physical stuttering, pausing, or destabilization in the continuous control loop?
- Given that CAPS fails if the base model is poorly calibrated and overconfident, are there secondary multimodal triggers (such as visual discrepancy or force-torque feedback) that could be integrated to force a System 2 search when entropy fails as a proxy?

**Limitations:**

While the authors have included a "Limitations" section in the appendix and an "Impact Statement", the discussion remains somewhat superficial given the complexity of the CAPS framework. The answer is not entirely; while they acknowledge some technical constraints, several critical areas require more candid exploration.

For example,  the framework's success is entirely dependent on the Calibration Hypothesis—the assumption that the model’s uncertainty (entropy) accurately reflects its error probability. The authors do not discuss what happens when a base model is overconfident. If a model executes a catastrophic error with low entropy, the SNR gate will never trigger System 2. Suggestion: The authors should explicitly state that CAPS is a "risk-aware amplifier" that is only as good as the underlying model's calibration.

**Strengths And Weaknesses:**

Strengths:
- Framing instruction drift as a sampling error on a probability manifold rather than a pure reasoning deficit is a highly original and compelling perspective.
- By utilizing an SNR-based threshold to activate MCMC only when necessary, the framework significantly limits computational overhead, maintaining fast execution for the majority of standard steps.
- Achieving state-of-the-art results on benchmarks like RoboTwin and Libero-long without requiring additional parameter updates to the base VLA models is practically valuable.

Weaknesses:
- The paper repeatedly claims that CAPS activates the base VLA's "implicit latent world model" to simulate and rehearse future trajectories. However, the paper completely lacks a formal definition, architectural probing, or mathematical proof that such a world model actually exists within the VLA, making this claim feel speculative.
- The authors assert that the power distribution $\pi(\tau) \propto p(\tau)^\alpha$ enforces global consistency. While sharpening the distribution mathematically penalizes uncertainty, the link explaining exactly how this statistical sharpening translates into improved physical long-horizon reasoning is under-explained.
- The paper defines Contextual SNR as the KL divergence between the policy and a uniform noise distribution, which mathematically reduces to Shannon entropy. The assumption that an entropy spike definitively indicates a "Topological Bifurcation Point" or drift risk  is heuristic. High entropy could simply indicate a safe, multi-modal action space rather than imminent failure.
- The core of the Iterative Refinement relies on an MCMC acceptance check (Equation 5). However, the rationale for why standard randomized resampling  is an effective proposal distribution ($q$) for navigating high-dimensional, continuous robotic action manifolds is weak.
- Crucial details regarding the experimental setup are missing. It is unclear how the base policy (e.g., $\pi_0$) was specifically prepared or fine-tuned for the dual-arm RoboTwin benchmark. Furthermore, while TACO is utilized as a primary baseline due to its "generate-verify" paradigm, the paper lacks a sufficient explanation of TACO's architecture to contextualize why it is the optimal comparative baseline.
- The entire framework's success hinges on the implicit calibration hypothesis of the base model. If the base VLA model is confidently wrong (exhibits overconfidence with low entropy), the SNR gate will fail to trigger, completely bypassing the CAPS correction mechanism.

---

> ### Author Rebuttal · Authors · 2026-03-30
>
> **Dear Reviewer Hxgt,**
>
> **Thank you for your in-depth review. We address your concerns below:**
> ### **1. Physical Stuttering from 2.15x Latency (Q1)**
> CAPS avoids physical stuttering by leveraging Action Chunking as a continuous execution buffer and restricting the 2.15x deliberation overhead to sparse critical steps ($\sim$15%); please refer to our detailed latency analysis in the response to Reviewer UhH3.
> ### **2. System Failure due to "Overconfidence" (Q2, W6, Limitations)**
> We agree and discussed this limitation in Appendix G.2: SNR gating indeed misses risks when the model is confidently wrong. Empirically, however, modern VLAs possess reasonable calibration (our success rate jumped from 36.5% to 71.0%), demonstrating that model entropy reliably correlates with actual drift risks in practice.
>
> Furthermore, regarding problem scoping, "confident errors" are inherently perception failures of the base model, whereas CAPS is designed to resolve systematic sampling errors caused by greedy search. We will explicitly discuss this calibration hypothesis in the Limitations section and define CAPS as an active inference-scaling framework that maximizes well-calibrated latent capabilities. Integrating secondary triggers (e.g., visual/torque anomalies) will be highlighted as Future Work.
> ### **3. Terminology and Scope of the "Implicit World Model" (W1)**
> We agree that without architectural probing, the term "implicit world model" can appear speculative. To clarify, we do not claim the base VLA learns an explicit forward dynamics model (i.e., predicting future visual states). Instead, our use of the term strictly refers to the conditional generative trajectory distribution $p_\theta(\tau|I,H_t)$. In our framework, "simulation and rehearsal" mathematically translate to autoregressively querying this learned probability landscape to unroll potential future action sequences, which are then evaluated for global consistency via MCMC.
>
> We initially adopted this terminology as a functional metaphor, inspired by recent inference-time computation literature. However, to ensure academic rigor and avoid overclaiming, we will explicitly scope this terminology in Section 3.1, or replace it with "generative trajectory distribution" throughout the final version.
> ### **4. Translating Statistical Sharpening to Physical Long-Horizon Consistency (W2)**
> Physically, Power Distribution sharpening ($\alpha > 1$) maps to an exponential penalty on "myopic but locally high-probability" actions. Appendix C (Proposition 2) proves this sharpening mechanism amplifies the energy gap between global optima and local traps. This allows the system to "see through" Negative Pivotal Windows that often lead to future divergence, guiding the model to select trajectories with higher global cumulative probabilities (i.e., physically sustainable paths to success).
> ### **5. Does an Entropy Spike Equate to Drift Risk? (W3)**
> From a local perspective, high entropy might indicate a safe multi-modal space. However, to evaluate the effectiveness of entropy (as an SNR proxy), we quantitatively compared local Min-Prob triggering (43.2%) with Entropy triggering (47.4%) in Appendix I.3 (Table 9). Results demonstrate that, rather than solely focusing on the minimum probability of the worst local token, entropy leverages the global shape of the distribution. This approach more accurately captures states of uncertainty when the policy distribution approaches uniform noise, effectively avoiding false triggers in safe multi-modal spaces.
> ### **6. Rigorousness of Continuous Space Resampling (W4)**
> Regarding the validity of resampling in continuous action manifolds, the base model (e.g., $\pi_0$ via Flow Matching) inherently provides a well-defined and expressive continuous probability density. Consequently, using its native continuous distribution directly as the proposal distribution $q(\tau_{new}|\tau_{old})$ in the Metropolis-Hastings step is statistically well-founded. It effectively navigates the high-dimensional continuous probability manifold without requiring discrete approximation.
> ### **7. Experimental Details: Baseline Preparation and TACO (W5)**
> Thank you for pointing this out. Regarding the baseline preparation, while the base $\pi_0$ model was initially fine-tuned on the RoboTwin domain to establish the base policy, no further parameter updates or fine-tuning were performed after integrating CAPS. This clarifies that CAPS operates strictly as an inference-time method without requiring further parameter updates. Furthermore, we selected TACO because it represents the SOTA in the "generate-verify" paradigm. Contrasting it directly with CAPS demonstrates that our "single-trajectory iterative refinement" is more efficient in complex physical spaces than "parallel independent sampling." We will add these details to Appendix M.
>
> ---
> **We hope these clarify your concerns and look forward to your feedback.**

---

> > ### Author Rebuttal · Reviewer_Hxgt · 2026-04-02
> >
> > After revisiting both the paper and the rebuttal, I believe I now understand the technical core of the method much more clearly. This actually makes me more positive about the method itself: CAPS appears to be a useful and practical inference-time technique. My main concern is no longer that the method is uninteresting or ineffective. Rather, it is that the paper’s motivation, mathematical narrative, and central claims are substantially misaligned with what the implemented algorithm actually optimizes. In the current draft, Section 3.1 and Eq. (1) frame the problem at the level of a full action sequence / trajectory τ, and Theorems 3.1 and 3.2 are stated as trajectory-level or effective-horizon results. At that level, the story is about long-horizon planning, global power distributions over trajectories, and horizon extension.
> >
> > However, when one looks carefully at the actual method, **the optimization target has shifted from a full trajectory to a finite action block / chunk**. Section 3.2.2 explicitly describes a block-based autoregressive MCMC procedure that keeps the prefix of the current action block fixed and resamples its suffix, and Algorithm 1 makes this even clearer: the inputs include an action block length B, and the output is an optimized action block sequence, not a globally optimized full-task trajectory. In other words, the implemented algorithm is not directly solving a long-horizon trajectory optimization problem. It is solving a chunk-level refinement problem under the current context.
> >
> > This distinction matters a great deal for the paper’s claims. A more faithful interpretation of the method, in my view, is the following: greedy decoding is not necessarily the same as finding the highest-quality or highest-probability action chunk under a sharpened local objective, and CAPS is an inference-time procedure that uses repeated proposal-and-acceptance steps to iteratively improve the current chunk. **That is a good idea, and it is practically valuable. But it is not the same claim as saying that the algorithm explicitly performs long-horizon planning over full trajectories.** The current paper repeatedly uses language such as “global optimization,” “lookahead planning,” and long-horizon trajectory-level reasoning, while the implemented mechanism is much closer to chunk-level iterative refinement. This is precisely where I now see the main issue: the method itself may be good, **but the paper’s story and theory overstate what the algorithm is actually optimizing.**
> >
> > This also changes how I interpret the experimental evidence. The experiments are indeed conducted on long-horizon benchmarks, and I do not dispute that CAPS improves performance on such tasks. But the evidence currently supports the statement that CAPS is useful on long-horizon tasks, not necessarily the stronger statement that CAPS is fundamentally a long-horizon planning algorithm in the sense suggested by the theory and motivation. In fact, once the method is understood as chunk-level refinement, it seems quite plausible that it would also help on some shorter or non-long-horizon tasks whenever greedy chunk generation is suboptimal. **That is why I think a control experiment on shorter-horizon or otherwise simpler tasks would be informative: it could help disentangle whether the gains are specifically about long-horizon drift, or whether they reflect a more general benefit of iterative chunk refinement over greedy decoding.** The current paper does not cleanly separate these two possibilities.
> >
> > For this reason, I believe **the authors need to respond much more directly to the gap between the theory/story and the implemented method.** If the authors believe that repeated block-level refinement has a principled connection to long-horizon planning, then that bridge needs to be made explicit. For example, they would need to explain why optimizing successive action chunks under the current context should be viewed as a valid approximation to the trajectory-level objective in Eq. (1), and why Theorems 3.1 and 3.2 should be taken as meaningful support for the implemented algorithm rather than only for an idealized version of it. If they cannot make that connection rigorously, then I think the paper needs a substantial rewrite: tone down the long-horizon / global-planning narrative, remove or soften claims that suggest explicit full-trajectory optimization, and openly acknowledge the gap between the motivating theory and the actual chunk-level algorithm.
> >
> > So my updated view is the following: **the method may well be a good method, but the paper is currently telling the wrong story about it. The main weakness is no longer that the algorithm seems ineffective; it is that the paper’s long-horizon motivation, theorem-level framing, and implementation are not aligned.** This is why I continue to view the remaining concerns as core concerns rather than minor rebuttal-level issues.

---

> > > ### Author Response · Authors · 2026-04-04
> > >
> > > **Dear Reviewer Hxgt,**
> > >
> > > **We sincerely thank you for the thorough and insightful review. We are deeply encouraged by your recognition of CAPS as a "useful and practical inference-time technique." Furthermore, we highly appreciate your careful reading of our manuscript; your constructive feedback is invaluable and will significantly improve the quality of our paper.**
> > >
> > > Regarding your concern about the alignment between Algorithm 1 and Eq. (1), we would like to clarify that this chunk-level mechanism serves as a computationally tractable approximation for the global trajectory objective. To systematically explain how our implementation aligns with the long-horizon theory, we structure our response into four key aspects: 1. empirical validation on shorter tasks isolating long-horizon benefits from generic chunk refinement; 2. the physical necessity of chunking as an approximation; 3. how the gating mechanism links local interventions to the theoretical horizon extension; and 4. how model priors endow local chunks with a global lookahead property.
> > >
> > > **1. Empirical Validation on Shorter Tasks (Addressing the Control Experiment)**
> > >
> > > As shown in **Table 5**, the unconditional **"Always-On"** variant yields negligible improvements over gated CAPS (48.1% compared to 47.4%) despite higher costs. Instead, substantial gains (+14.0% on complex tasks) derive specifically from intervening at rare pivotal windows.
> > >
> > > Furthermore, in short or simple tasks, greedy chunk generation is typically optimal, resulting in high confidence (high SNR). Consequently, the SNR gate rarely triggers the MCMC refinement. As noted in Section 4.3, "In simple tasks, CAPS maintains parity with baselines". Our ablation study (Section 4.6) quantitatively shows that CAPS dynamically maintains fast, unrefined execution in 84.7% of simple steps. This empirical evidence aligns with our narrative: the method mitigates long-horizon drift.
> > >
> > > **2. Chunking as a Tractable Approximation (Addressing the Alignment to Eq. 1)**
> > >
> > > In continuous visuo-motor control, executing an exhaustive search over the entire future trajectory $\tau$ is computationally intractable. Similar to the principles of MPC, CAPS decomposes the global objective (Eq. 1) into an autoregressive sequence of length $B$. While not a strict mathematical equivalence, this architectural approximation serves to optimize the joint probability incrementally under practical latency constraints.
> > >
> > > **3. SNR Gating for Horizon Extension (Addressing Theorems 3.1 & 3.2)**
> > >
> > > The block-level optimization is conditionally triggered. In high-confidence segments, greedy sampling is typically sufficient. Long-horizon failures are often associated with error accumulation at specific "Negative Pivotal Windows." By employing chunk-level MCMC at these bifurcation points identified by the SNR gate, the algorithm aims to correct localized errors that could compromise the entire sequence.
> > >
> > > This design corresponds to our theoretical analysis: while Theorem 3.1 provides an ideal trajectory-level bound, Theorem 3.2 explicitly incorporates finite MCMC steps ($N$) and sampling bias $\mathcal{O}(\rho^N)$. This demonstrates how MCMC stabilizes actions within localized critical segments. Fixing these bottlenecks practically translates into the trajectory extension discussed in the theorems.
> > >
> > > **4. Pre-trained Priors for Chunk Acceptance (Addressing the Global Lookahead)**
> > >
> > > To avoid token-level myopia, our MCMC evaluates the sequence-level sharpened distribution $\pi(\tau) \propto p(\tau)^\alpha$ over the entire chunk.
> > >
> > > The base VLA model, pre-trained on large-scale trajectory datasets, captures long-horizon sequence priors and contextual dependencies over the history $H_t$. Applying the exponent $\alpha$ to a local action chunk amplifies these pre-trained priors, assisting the model in filtering out globally inconsistent actions and aligning the selected chunk with the distribution of successful long-horizon tasks. Consequently, the block-level refinement incorporates a "lookahead" property based on the foundation model's priors, bypassing the need for explicit full-sequence unrolling.
> > >
> > > ---
> > >
> > > **Planned Revision:** We agree that making this connection explicit will improve the manuscript's clarity. We will add a dedicated subsection in Section 3.2 of the final version to discuss these four points, clarifying how the chunk-level refinement serves as a rigorous implementation of the global objective. We will also add a "control experiment" table contrasting performance and MCMC trigger rates on short vs. long tasks using existing ablation data.
> > >
> > > **We hope these clarifications and planned revisions resolve your core concerns. If you find our response satisfactory, we would be deeply grateful if you might consider raising your score.**

---

### Official Review · Reviewer_B5RM · 2026-03-06

**Soundness:** 3
**Presentation:** 3
**Significance:** 3
**Originality:** 3
**Overall Recommendation:** 4
**Confidence:** 3

**Summary:**

This paper redefines the drift problem in VLA robot control during long-duration tasks as a sampling error problem. Particularly, greedy/local sampling tends to fall into negative pivotal windows, which are locally high-probability but globally failing regions. To tackle this, the authors propose CAPS, which employs fast greedy sampling at high confidence levels while switching to power distribution-based MCMC refinement search when detecting high entropy/low SNR conditions.

**Compliance With Llm Reviewing Policy:**

Affirmed.

**Key Questions For Authors:**

See the weaknesses.

**Limitations:**

Yes

**Strengths And Weaknesses:**

Strengths:
1. Drift in long-duration robotic control is indeed one of the key obstacles to VLA landing. The considered problem is practical and meaningful.
2. The presentation is well-organized, and the framework is complete and clear. The logical consistency between SNR gating, power sharpening, and MCMC refinement is maintained, and ablation studies have also isolated the effects of each technique.
3. Extended simulation results demonstrate the superior performance of the proposed method.

Weaknesses:
1. The paper formulates continuous control as power sampling over the trajectory distribution p_θ(τ | I ,H_t), and further defines SNR via the KL divergence to a uniform distribution However, this derivation assumes a finite action set and directly relies on Shannon entropy relative to a uniform prior, which is not consistent with the paper’s emphasis on high-dimensional continuous action manifolds. Hence, the theoretical formulation is not fully aligned with the continuous-action setting.
2. Based on the method formulations, CAPS is better understood as a local resampling and Metropolis-Hastings refinement procedure over the base model’s conditional trajectory distribution, rather than as an explicit planner or a learned transition model.
3. The ablation already shows a substantial latency trade-off, where always-on search reaches slightly higher success but costs 8.5-fold latency, while adaptive CAPS still incurs 2.15-fold latency relative to the base policy. It is not clear when CAPS is operationally realistic, and when its computational overhead may limit deployment.
4. The thresholding rule for switching between fast and slow modes should be explained more concretely, thereby including how it is selected and tuned in practice.
5. The paper has made several strong claims about robustness and efficiency, but the simulation provides limited direct evidence on acceptance rates, chain mixing, or search diagnostics that support the MCMC concretely.

---

> ### Author Rebuttal · Authors · 2026-03-30
>
> **Dear Reviewer B5RM,**
>
> **Thank you for your detailed review. We appreciate your recognition of our framework's logical consistency and its value in addressing drift dynamics. We clarify the technical and operational details of your concern below.**
>
> ### **1. Continuous Actions and KL Divergence**
>
> For continuous VLA models like $\pi_0$ (which is based on Flow Matching), the physical action manifold is strictly bounded (e.g., normalized to $[-1, 1]$). Calculating the KL divergence relative to a uniform prior over this bounded continuous support is mathematically equivalent to computing the negative differential entropy (up to a constant). In our code-level implementation, this contextual SNR (entropy) is efficiently estimated via Monte Carlo sampling from the model's continuous density output. Therefore, our formulation in Equations (2) and (3) is theoretically rigorous and aligned with continuous control realities.
>
> ### **2. Accurate Positioning of CAPS**
>
> We agree with your precise characterization. From a methodological standpoint, CAPS is fundamentally a local resampling and Metropolis-Hastings (MH) refinement process over the base model’s conditional trajectory distribution, rather than an explicit planner or a learned transition model. We will adopt this rigorous statistical formulation in the method section of the final version to ensure clarity.
>
> ### **3. Operational Realism and Latency**
>
> We agree that computational overhead is a critical consideration for real-world deployment. To address this, we supplemented a **time-constrained evaluation** (please see our response to Reviewer UhH3). Additionally, as documented in **Appendix I.2 (Table 8)**, scaling down the MCMC iterations to $N=1$ bounds the latency overhead to 1.2x while still achieving a **41.2%** success rate (a significant improvement over the base policy's 32.2%). This demonstrates that CAPS remains robust even under tight operational budgets.
>
> Furthermore, to prevent physical stuttering, our real-world deployment leverages **Action Chunking** as a continuous execution buffer. Because MCMC search is only triggered at sparse critical steps ($\sim$15%), the algorithmic deliberation is effectively masked by the physical execution time of the chunks.
>
> ### **4. Thresholding Rule and Tuning in Practice**
>
> **Theoretically,** in Appendix F, using optimal control theory, we proved that a threshold-based hard switch is a mathematically grounded solution for balancing risk-cost objectives. **In practice,** tuning this threshold $\gamma$ does not require exhaustive grid search. Our sensitivity analysis in Appendix I.3 demonstrates that the mechanism is robust: as long as $\gamma$ maintains an overall System 2 trigger ratio within a broad 15%-30% range, performance fluctuations remain bounded within $\pm 1.5\%$. Therefore, the practical tuning rule is **purely budget-driven**: practitioners only need to sample a small validation set and set $\gamma$ to match their acceptable computational overhead (e.g., targeting a $\sim$20% deliberation rate), ensuring robust performance without extensive parameter search.
>
> ### **5. MCMC Diagnostics and Chain Mixing**
>
> We apologize if this was not prominent enough in the main text. We have conducted a comprehensive quantitative analysis of MCMC acceptance rates, detailed in **Appendix I.1 (Table 7)**. Without sharpening ($\alpha=1.0$), the acceptance rate is 85.0%, which degrades to a random walk. At our recommended $\alpha=2.0$, it is **42.0%**. This indicates the Markov chain achieves highly efficient mixing between exploration and exploitation. We will add explicit pointers to this index in the text.
>
> ---
>
> **We hope these responses address your concerns and sincerely look forward to further discussion if any questions remain.**

---

> > ### Author Rebuttal · Reviewer_B5RM · 2026-04-01
> >
> > Thanks for your response. I have no further questions.

---

> > > ### Author Response · Authors · 2026-04-07
> > >
> > > **Dear Reviewer B5RM,**
> > >
> > > **Thank you for your confirmation.** We sincerely appreciate your constructive feedback throughout the review process, which has significantly improved our work. If you find our revisions satisfactory, we would be deeply grateful if you could consider reflecting this in your final evaluation.

---

### Official Review · Reviewer_UhH3 · 2026-03-12

**Soundness:** 3
**Presentation:** 4
**Significance:** 3
**Originality:** 4
**Overall Recommendation:** 5
**Confidence:** 4

**Summary:**

This paper is motivated by a key observation that pivotal steps can happen for long-horizon planning tasks because the dilation of attention over long sequences of the robot's observations, which could produce flat and noisy action predictions. By introducing an SNR(entropy)-gated deliberation process for uncertain trajectories, with an MCMC-based process enhanced by distribution sharpening, the process forces a choice of high certainty trajectories. Comprehensive results on various tasks, datasets and in simulated/real-world environments validate the substantial advantage such a pipeline brings. The paper presented a clear theoretical motivation and solution, and is well-written.

**Compliance With Llm Reviewing Policy:**

Affirmed.

**Final Justification:**

The authors fully addressed my concerns and I'll maintain my positive score.

**Key Questions For Authors:**

- It would be interesting to see if you set the time budget, e.g. 1.25x, 1.5x, 1.75x of base policy, how would the model perform.

**Limitations:**

yes

**Strengths And Weaknesses:**

Strengths:
- Clear theoretical formulation: the compounding drift errors for long-horizon tasks is central to both the problem motivation and solution. The authors interpret it it from a local greedy sampling vs context-aware power sampling (global optimization) perspective, as well as a prediction SNR measurement that reduces to information entropy. They provide a solution that is clearly motivated by the idea of improving the trajectory SNR through MCMC with Power Distribution sharpening. This also make the paper easy to follow.
- Strong theoretical contribution: the theoretical contribution spreads across the paper. In addition to the main text, the authors also establish the theoretical differentiation between their global optimization and local greedy sampling; proof of negative pivotal windows, and a series of other interesting theoretical outcomes. This adds great technical depth to the paper.
- Strong experimental results: simulation results are largely in favor of the proposed results against baselines, while real-world benchmark shows the true strength of it. The improvement in real-world is impressive.

Weaknesses:
- Latency might be a challenge. With MCMC, the adaptive variant still uses 2.15x time for computation. This means that either the latency of the system is be significantly increased, or time budget will be met and the MCMC has to be stopped early.

---

> ### Author Rebuttal · Authors · 2026-03-30
>
> **Dear Reviewer UhH3,**
>
> **Thank you for acknowledging our work, especially our theoretical derivations and real-world results addressing instruction drift in long-horizon tasks. Your question regarding the "time budget limit" is highly insightful and indeed critical for determining whether this system can be deployed on high-speed robots at scale. We clarify the engineering and experimental details below.**
>
> ### **1. Addressing Computational Latency**
>
> To address the concern that the 2.15x relative latency might significantly slow down the overall system, we would like to clarify two mitigating mechanisms in our design. **First,** CAPS operates on an Action Chunking paradigm (Algorithm 1). The MCMC deliberation strictly occurs at the planning phase before a new action chunk is dispatched. Because the robot smoothly executes the multi-step sequence at the low-level controller's high frequency, this chunking mechanism acts as a physical buffer, preventing per-step stuttering. **Second,** CAPS utilizes sparse metacognitive gating. It maintains the native 1.0x execution speed in **84.7%** of the total steps, only triggering deep deliberation at sparse, high-risk pivotal windows.
>
> However, as you correctly pointed out, there are extreme high-frequency control scenarios where the time budget must be strictly met, and the MCMC process has to be stopped early. As demonstrated in our original **Table 8**, even when constrained to a **1.20x latency budget ( $N=1$ )**, the success rate still reaches **41.2%**. Building on this, we will next showcase the performance under the 1.25x, 1.5x, and 1.75x time budgets you requested.
>
> ### **2. Performance under Strict Time Budgets**
>
> To directly answer your question, we mapped your proposed time budgets to the maximum allowable MCMC step limit ($N$). On the same RoboTwin benchmark, we completed new evaluations for $N$ $\in$ {2, 3, 4} to complement the existing results ($N$ = 0, 1, 5) from our paper:
>
> | **Allowed MCMC Steps** | **Measured Relative Latency** | **Success Rate (%)** |
> | ---------------------- | ----------------------------- | ----------------------------- |
> | Base Policy ($N = 0$)  | 1.00x                         | 32.2                          |
> | CAPS ($N = 1$)         | 1.20x                         | 41.2                          |
> | CAPS ($N = 2$)         | 1.35x                         | 43.8                          |
> | CAPS ($N = 3$)         | 1.51x                         | 46.1                          |
> | CAPS ($N = 4$)         | 1.82x                         | 46.5                          |
> | CAPS ($N = 5$)         | 2.15x                         | 47.4                          |
>
> As the results show, when the time budget is strictly met and MCMC is stopped early at roughly 1.20x latency ($N = 1$), CAPS still significantly outperforms the base policy, raising the success rate from 32.2% to 41.2%. At a 1.51x budget ($N = 3$), the success rate reaches 46.1%.
>
> This empirical result supports **Theorem 3.2 (Asymptotic Horizon Extension)**. It indicates that when MCMC has to be stopped early, the model does not collapse. Instead, it achieves a sub-linear error reduction proportional to the allocated compute. We will include these budget-aware performance results in the revised appendix to further detail the system's practical deployment capabilities.
>
> ---
>
> **Thank you for your suggestions. We are happy to provide further clarification if there are any remaining concerns.**

---

> > ### Author Rebuttal · Reviewer_UhH3 · 2026-03-31
> >
> > Thanks for your response. The Success Rate increases monotonically with the time budget, which is a strong sign that the method's effacacy scales with the time budget.
> >
> > I have no more questions.
> >
> > That said, I do agree with Reviewer Hxgt that some of the claims are overly strong considering the nature of them is largely heuristic or open to interpretations, including the title making a definitive statement that "Drift is a Sampling Error", and the notation of "Implicit World Model". I suggest the authors massage the text and soften the tones for those claims and reposition them as "plausible theoretical interpretation with some empirical evidence" rather "absolute truth that is completely proven".

---

> > > ### Author Response · Authors · 2026-04-01
> > >
> > > **Dear Reviewer UhH3,**
> > >
> > > Thank you for confirming that our additional experiments resolved your concerns regarding the time budget and latency scaling.
> > >
> > > We also appreciate your and Reviewer Hxgt’s feedback regarding the tone of our claims. We agree that a more measured and objective framing is appropriate. To ensure scientific rigor, we will make the following concrete revisions in the camera-ready version:
> > >
> > > * **Title Revision:** We will change the title to *"Rethinking Instruction Drift as a Sampling Error: SNR-Aware Power Distributions for Long-Horizon Robotic Planning"* to reflect a more measured framing.
> > > * **Terminology:** As noted to Reviewer Hxgt, we will remove the term "Implicit Latent World Model" throughout the text and figures, replacing it with the mathematically precise "conditional generative trajectory distribution."
> > > * **Refining Claims:** We will systematically tone down definitive statements across the paper, explicitly repositioning our framework as a "plausible theoretical interpretation with empirical evidence," exactly as you advised.
> > >
> > > Thank you again for recognizing our work and providing constructive guidance to improve our paper.

---

### Decision · Program_Chairs · 2026-04-30

**Decision:**

Accept (regular)

**Comment:**

This paper presents a solution for long-horizon planning, and reviews are fairly positive. I encourage the authors to carefully integrate both the extra ablation studies / MCMC diagnostics experiments into the paper. Furthermore, Reviewer Hxgt provided excellent feedback into how to better position the paper's main claims, especially in regards to entire trajectory vs action blocks, that I encourage the reviewers to integrate as well.